# Sensitivity of Biogenic Volatile Organic Compound Emissions to Leaf Area Index and Land Cover in Beijing

Hui Wang[1, 4], Qizhong Wu[1, 4], Hongjun Liu[2], Yuanlin Wang[3, 4], Huaqiong Cheng[1, 4], Rongrong Wang[1, 4], Lanning Wang [1, 4], Han Xiao[1, 4], Xiaochun Yang[1, 4]

[1]College of Global Change and Earth System Science, Beijing Normal University, Beijing 100875, China
[2]Department of Physics, Beijing Normal University, Beijing 100875, China
[3]State Key Laboratory of Atmospheric Boundary Layer Physics and Atmospheric Chemistry, Institute of Atmospheric Physics, Chinese Academy of Sciences, Beijing 100029, China
[4]Joint Center for Global Changes Studies, Beijing Normal University, Beijing 100875, China

*Correspondence to*: Qizhong Wu (wqizhong@bnu.edu.cn)

**Abstract.** The Beijing area has suffered from severe air pollution in recent years, including ozone pollution in the summer. In addition to the anthropogenic emissions inventory, understanding local ozone pollution requires a reliable biogenic volatile organic compound (BVOC) emission inventory. Forest coverage rose from 20.6% to 35.8% from 1998 to 2013 in Beijing according to the National Forest Resource Survey (NFRS), and accurate representations of land cover for recent years is crucial for estimating BVOC emissions and their impacts on air quality. In this study, we established a high resolution BVOC emission inventory in Beijing using the Model of Emission of Gases and Aerosols from Nature (MEGAN) v2.1 with three independent leaf area index (LAI) products and three independent land cover products. Various combinations of the Global LAnd Surface Satellite (GLASS), Moderate-Resolution Imaging Spectroradiometer (MODIS) MCD15, and GEOland (GEO) v2 LAI datasets and the Finer Resolution Observation and Monitoring of Global Land Cover (FROM-GLC), MODIS MCD12Q1 PFT products, and Climate Change Initiative Land Cover (CCI-LC) products are used in five model sensitivity experiments (E1-E5), and the experiment using the FROM-GLC with highest spatial resolution of 30m and GLASS LAI products was treated as the baseline. These sensitivity calculations were driven by hourly, 3 km meteorological fields from the Weather Research and Forecasting (WRF) model. The following results were obtained: (1) According to the baseline estimate, the total amount of BVOC emissions is 75.9 Gg for the Beijing area, and isoprene, monoterpenes, sesquiterpenes and other VOCs account for 37.6%, 14.6%, 1.8%, and 46% of the total, respectively. Approximately three-quarters of BVOC emissions occur in the summer. (2) According to the sensitivity experiments, the LAI input does not significantly affect the BVOC emissions. Using MODIS MCD15Q1 and GEO v2 LAI led to slight declines of 2.6% and 1.4%, respectively, of BVOC emissions in the same area. (3) The spatial distribution of PFTs from different inputs strongly influenced the spatial distribution of BVOC emissions. Furthermore, the cross-walking table for converting land cover classes to PFTs also has a strong impact on BVOC emissions; the sensitivity experiments showed that the estimate of BVOC emissions by CCI-LC ranged from 42.1 to 70.2 Gg depending on the cross-walking table used. Adopting the "maximum biomass" table made the CCI-LC relatively consistent with the other two LC products, such that the estimates of BVOC emissions in the Beijing region by the three LC products consistently fell

within the range of 28.5–30.5 Gg for isoprene, 9.3–10.1 Gg for monoterpenes, 1.2–1.4 Gg for sesquiterpenes, and 28.3–35.6 Gg for other BVOCs.

## 1. Introduction

Biogenic volatile organic compounds (BVOCs) play a significant role in the atmospheric environment because of the large quantities emitted and their high reactivity (Fuentes et al., 2000;Guenther et al., 1995). BVOCs can form secondary organic aerosol (SOA) (Claeys et al., 2004;Kavouras et al., 1998) as well as affect tropospheric ozone ($O_3$) and nitrogen oxides radicals ($NO_X$) (Fuentes et al., 2000;Seinfeld and Pandis, 2012). BVOC emissions are affected by meteorological conditions, including solar radiation,

temperature, and concentration of carbon dioxide (Arneth et al., 2007;Guenther et al., 2006;Guenther, 1993). Furthermore, the changing climate will lead to changes in environmental conditions and anthropogenic activities, which have an impact on the BVOC emissions, leading to feedback effects on climate and human beings (Penuelas and Staudt, 2010). Under the influence of global warming, as estimated by Stavrakou et al. (2014), the isoprene emissions in East Asia and China increased by 0.2%

$yr^{-1}$ and 0.5% $yr^{-1}$, respectively, from 1979 to 2005. In addition, considering the severe air quality problem in China, addressing the contribution from natural and anthropogenic sources could benefit attempts to improve air quality.

Klinger et al. (2002) estimated the total amount of BVOC emissions in China at about 21.0 Tg, with terpenoids accounting for approximately 8.06 Tg. That terpenoid emission estimate is similar to the 10.9

20    Tg estimated by Tie et al. (2006). Since subtropical regions in China have severe ozone pollution and abundant forests, multiple studies have focused on local biogenic emissions and their potential effects on urban air quality (Wang et al., 2011;Leung et al., 2010;Wang et al., 2013). As a typical city in North China, Beijing faces severe ozone pollution in summer (Wang et al., 2006;Safieddine et al., 2016;Zhao et al., 2010). Furthermore, model and satellite results both indicate that the North China plain suffers

relatively more severe $O_3$ pollution than southern regions in China during summer (Zhao et al., 2010;Safieddine et al., 2016). Because Beijing is surrounded temperate forests, it is necessary to consider the influence from biogenic emissions on local air pollution. Previous studies have carried out some calculations of local BVOC emissions (Wang et al., 2003;Klinger et al., 2002), but these estimates were made for an earlier period (1998–2000), and the China Forestry Database (http://data.forestry.gov.cn/lysjk)

provided by the National Forest Resources Surveys shows that the forest coverage rate in Beijing rose

from 20.6% to 35.8% from 1998 to 2013. Furthermore, the dominant tree species of the local forests are *Quercus* and *Populus*, which are strong isoprene emitters (Wang et al., 2003). Therefore, an up-to-date BVOC emission inventory is needed to evaluate biogenic effects on local air quality.

The Model of Emission of Gases and Aerosols from Nature (MEGAN) (Guenther, 2006;Guenther et al., 2012) model is the most commonly used BVOC emission model and has been widely used in air quality simulations in China (Gao et al., 2016;Geng et al., 2011;Fu and Liao, 2012;Wang et al., 2018). In this study, we adopted the MEGAN model to estimate BVOC emissions across the Beijing region. The Leaf Area Index (LAI) and land cover (LC) are two important factors for the BVOC emission estimates. The LAI and LC determine the biomass and BVOC emission potential, respectively, in the MEGAN model. Therefore, multiple satellite-based LAI and LC products were examined in this study to investigate the uncertainties associated with the LAI and LC inputs. Sec. 2 introduces the models and datasets used in this study. In Sec. 3, we elaborate our BVOC emission inventory as well as the sensitivities of the model to different satellite inputs. We present our conclusions in Sec. 4.

## 2. Methodology

### 2.1 Model description

The BVOC emission rate in MEGAN is calculated as follows (Guenther et al., 2006):

$$Emission = [\varepsilon] \cdot [\gamma] \cdot [\rho]$$

where $\varepsilon$ is a factor that accounts for the emission rates of different compounds under standard canopy conditions, $\gamma$ is an activity factor that accounts for the environmental variance, and $\rho$ is a factor that accounts for chemical and physical losses in the plant canopy layer.

#### 2.1.1  Emission factors

The standard emission rates in MEGAN adopt the canopy-scale emission factors ($\mu g\ m^{-2}\ h^{-1}$), which are converted from the measured leaf/branch-scale emission factors ($\mu g\ g^{-1}\ h^{-1}$). The leaf/branch-scale emission factor, leaf mass per area (LMA, $g\ m^{-2}$), and the standard environmental factor were used to convert the leaf/branch-scale emission factors to canopy-scale emission factors (Leung et al., 2010;Guenther et al., 2006). In MEGAN, the $\varepsilon$ can be described by either the specific tree species map or the plant functional type (PFT) map (Guenther et al., 2006). In this study, we adopted six PFTs to explain the standard emission factors: broadleaf trees, needleleaf trees, shrubs, grass, corn and other crops. Moreover, we modified the standard emission factor of isoprene for every PFT according to local field measurements from previous publications (Wang et al., 2003;Klinger et al., 2002). Based on the area data

of diverse tree species from the eighth NFRS (Table S1) which came from Ren et al. (2017), the emission factors of isoprene for the PFTs mentioned above are calculated as

$$\varepsilon_j = \sum \varphi_i LMA_i \frac{s_i}{s_j}$$

where $\varepsilon_j$ is the canopy-scale emission factor of the PFT species $j$, $\varphi_i$ is the leaf-scale emission factor for the vegetation species $i$, and the $S_i/S_j$ represents the area proportion of vegetation species $i$ in PFT $j$.

Due to the lack of measurements of emission factors in the local area, the default emission factors of MEGAN v2.1 were used for all other VOCs. Table 1 shows the original and adjusted standard emission factors of isoprene.

### 2.1.2 Environmental activity factor

The environmental activity factor accounts for the effects of leaf age, canopy meteorological environment, and soil moisture in MEGAN, and it can be expressed as:

$$\gamma = \gamma_{CE} \cdot \gamma_{age} \cdot \gamma_{SM}$$

where $\gamma_{CE}$ accounts for variations in the canopy meteorological conditions, $\gamma_{age}$ describes the effect of leaf age, and $\gamma_{SM}$ accounts for the impact of soil moisture.

Temperature and photosynthetically active radiation (PAR) are the main canopy meteorological variables that affect BVOC emissions. The emission of isoprene was modeled as fully light- and temperature-dependent according to the algorithms of isoprene emission conditions described by Guenther et al. (2012). Approximately 10% of monoterpenes and 50% of sesquiterpenes were treated as light- and temperature-dependent, and the others were treated as temperature-dependent species (Sakulyanontvittaya et al., 2008). The moisture factor $\gamma_{SM}$ was only considered for the isoprene emission. The corresponding canopy models were adopted to calculate the sunlit and shaded leaves temperature and light scattering of each PFT (Guenther et al., 1999;Guenther et al., 2006). The details of the algorithms used for isoprene and monoterpenes can be found in Guenther et al. (2012) and Sakulyanontvittaya et al. (2008).

### 2.2 Data & simulation description

In this study, we used a mesoscale weather model to provide the meteorological conditions. Due to the lack of direct observations of canopy emissions, we varied the input conditions to investigate the sensitivity of the simulation to the emission inventory. Three PFT and three LAI datasets were adopted to investigate the impact of these inputs.

### 2.2.1 WRF meteorological simulation

The Weather Research and Forecasting (WRF) V3.3.1 (Skamarock et al., 2008) model was used to provide the meteorological conditions. The initialization field and boundary conditions for WRF are provided by the National Centers for Environmental Prediction (NCEP) FNL (Final) Operational Global Analysis data (National Centers for Environmental Prediction, 2000) (https://rda.ucar.edu/datasets/ds083.2/). The boundary conditions are updated every 6 hours. The model domain contained three horizontally nested grids with the spatial resolution of 27-9-3 km and 31 vertical layers, including 4 layers of soil from Noah land Surface model (Tewari et al., 2004). The 3 km horizontal grid covered Beijing and was used for the BVOC emission inventory. The WRF model was initialized at 12:00 UTC, and the first 12 hours were spin-up time. The data of the period from 00:00 A.M. UTC to 23:00 P.M. UTC in the second day was cut and merged to estimate the BVOC emissions. The merged file was processed by the Meteorology-Chemistry Interface Processor (MCIP) (Otte and Pleim, 2010) tool to provide meteorological conditions for MEGAN model. The physical options used for the WRF model are presented in Table S2 in the supplement.

The daily temperature (T2) simulated by WRF was primarily verified by in-situ data from 15 monitoring sites among the Beijing region, and the daily downward shortwave radiation (DSW) was also validated using the in-situ data from Beijing Station. Figure 1 shows the time-series of stations-averaged daily T2 and DSW in Beijing Station. The mean error (ME), mean bias (MB), correlation coefficient (r), and root mean square error (RMSE) of stations-averaged T2 series are 1.76 °C, –1.42 °C, 0.99 and 2.13 °C, respectively. The high r (0.99) indicates that the simulation accurately reproduces the daily temperature variations. The ME, MB, r and RMSE of the DSW series are 71.06 W m$^{-2}$, 56.79 W m$^{-2}$, 0.51 and 102.17 W m$^{-2}$. The detail statistics factors of daily T2 among stations are given in Table 2, furthermore, the detail statistics factors of hourly T2 among stations are also given in Table S3 in the supplement for checking. Across all sites, Tong Zhou, Da Xing, and Fang Shan have the most obvious underestimates of surface temperature, with high negative biases of –4.84 °C, –5.1 °C 1, and –4.5 °C, respectively. Figure 2 shows the location of all sites, with the MB of T2 indicated by the color scale, and these sites are located in the suburban regions of Beijing, where are under the fast urbanization and BVOC emissions are lower, and the WRF didn't simulate the urban heat island phenomenon in these regions. And the main source of BVOCs is the rural forest around Beijing, and Table 2 as well as Figure 2 indicate relative good simulation among the sites in rural region; thereby, the simulation bias of the suburban regions can be expected to have little impact on the estimate of whole BVOC emissions.

### 2.2.2 Satellite datasets

The PFT and LC datasets include the Finer Resolution Observation and Monitoring of Global Land Cover (FROM-GLC) (Gong et al., 2013;Yu et al., 2014), the Moderate-Resolution Imaging Spectroradiometer (MODIS) MCD12Q1 PFT products (Friedl et al., 2010), and the Climate Change Initiative Land Cover (CCI-LC) products (ESA, 2017). Three LAI data products are adopted as the LAI input, including the Global LAnd Surface Satellite (GLASS) (Xiao et al., 2014;Xiao et al., 2016), MODIS MCD15A2 Version 5 (Knyazikhin et al., 1999), and Geoland (GEO) v2 (Baret et al., 2013;Verger et al., 2014b) LAI products. The LC datasets were regrided to the WRF grid by calculating the area fraction of each PFT, and the LAI datasets were converted from original grids to WRF grids by calculating the area mean LAI in the WRF grids.

The FROM-GLC is the first global LC product with 30 m spatial resolution (Gong et al., 2013). It is based on Thematic Mapper (TM)/ Enhanced Thematic Mapper (ETM) images and uses images from MODIS and Google Earth as references. Because the higher spatial resolution captures a more detailed distribution of PFTs, we used the latest version FROM-GLC-AGG (Yu et al., 2014) as the baseline PFT input. The spatial resolutions of the other two global LC products used to study the impact of the PFTs inputs, MODIS MCD12Q1 and CCI-LC, are 500 m and 300 m, respectively. The benchmark years of FROM, MODIS, and CCI-LC are 2010, 2013, and 2013, respectively. Since the forests would not obviously changed in three years, the FROM PFT were used to calculate the BVOC emissions with other inputs were for year 2013. The PFT map layer of MODIS MCD12Q1 product was directly used. CCI-LC uses the default cross-walking table given by Poulter et al. (2015) to convert the LC class maps to PFT maps. The FROM-GLC conversion process used the class legend description, with each LC type being classified into the corresponding PFT. The PFT proportions of the three LC products are presented in Table 3. All three products have similar percentages of needleleaf trees, but different percentages of broadleaf trees. The CCI-LC has lower broadleaf tree coverage compared to the other two products due to the impact of the cross-walking table. Figure 3 shows the spatial distribution of the four main PFTs in the model grids of the LC products. As shown in Figure 3, the three datasets have similar distributions, but differ in forest density. Because of the high emission factors of broadleaf trees, the highest broadleaf tree density of the MODIS LC data implies the highest emission density. Considering the high biomass and emission factors, the local broadleaf trees could lead to a considerable emission potential. In contrast, the distributions of the shrub and grass PFTs show higher variability than the tree PFTs, but the low emission factors limit their impacts on the estimate of terpenoid emissions.

Three different LAI datasets were adopted in this study: the GLASS v1.1, MODIS MCD15A2, and GEO v2 LAI products. All three datasets have a spatial resolution of 1 km. The temporal resolutions of GLASS and MODIS are both 8 d, and that of GEO v2 is 10 d. The GLASS v1.1 LAI products are retrieved from reprocessed Advanced Very High Resolution Radiometer (AVHRR) and MODIS reflectance data using the General Regression Neural Network (GRNN) (Xiao et al., 2014;Xiao et al., 2016), which was trained by the fused LAI from the MODIS and CYCLOPES LAI products. The GEO v1 adopts the Neural Network trained by the MODIS and CYCLOPES fused LAI to derive the LAI from the reflectance data from the SPOT/VEGETATION sensor (Baret et al., 2013). Based on GEO v1, the later GEO v2 employs a filtering approach to eliminate the outliers as well as Savitzky-Golay and climatology temporal smoothing and gap filling methods to ensure consistency and continuity (Verger et al., 2014a). Due to the diverse satellite data sources and algorithms, these three datasets are treated as dependent datasets and were used to study the impact of different satellite LAI inputs. Figure 4 (a)-(c) shows the spatial distribution of the three LAI products in the model grid in summer. Since the MODIS MCD15A2 uses the vegetation canopy radiation model to produce LAI products (Knyazikhin et al., 1999), the region assigned as non-pure vegetation types leads to missing values in the MODIS MCD12 Q1 LC products. Thus, the MODIS MCD15A2 LAI product has a bigger mask area in suburban areas and near water, which could lead directly to the loss of BVOC emissions in these areas. Figure 5 shows the monthly average LAI values of trees and grasses of the three products based on the MODIS MCD12Q1 LC. Only the grid cells in the region over which the MODIS MCD15A2 LAI is valid were taken into account. According to Figure 5, the three LAI products have nearly the same trend of LAI values for trees, and GEO v2 product has the highest LAI from May to September. The MODIS MCD15A2 product has the lowest LAI of the three products for the tree and herb vegetation. The peak LAI of trees occurs in July for all three products, and the mean LAI of the three products during the winter seasons are all below 1 because of the low biomass of local deciduous tree species. The direct validations by Xiao et al. (2016) showed that of the three products, the GLASS LAI is most consistent with observations. Therefore, we treated the GLASS LAI as the most accurate LAI and it was used in the baseline experiments.

Table 4 presents the configurations of the simulation experiments. The baseline experiment (E1) used the FROM PFT and GLASS LAI as inputs. Experiments E1-E3 used the same PFT input and varied the LAI inputs to investigate the impacts of the different LAI inputs. The effect of different PFT inputs was investigated in experiments E1, E4 and E5, which all used the same GLASS LAI input but used different LC datasets.

## 3. Results and description

MEGAN v2.1 can output 20 basic compounds, which can be divided into 150 VOC species (Guenther et al., 2012). In this study, the VOC species are divided into four groups: isoprene, monoterpenes, sesquiterpenes, and other VOCs. Monoterpenes include Myrcene, Sabinene, Limonene, 3-Carene, α-β-Ocimene, β-Pinene, α-Pinene, and other monoterpenes, and the sesquiterpenes include α-Farnesene, β-Caryophyllene, and other sesquiterpenes. The following sections are largely focused on the terpenoids because of their high reactivity and because they are better understood than the other VOCs, which are associated with larger uncertainties.

### 3.1 Quantity of BVOC emissions

According to the baseline experiment (E1), the quantity of BVOCs emitted annually in Beijing is 75.9 Gg; isoprene, monoterpenes, sesquiterpenes and other VOCs make up 37.6%, 14.6%, 1.8%, and 46.0% of the total, respectively. Table 5 presents the annual emission results of all experiments. E2 and E4 have similar total emissions at 75.7 Gg and 76.5 Gg, respectively, while E3 and E5 have lower emissions at 61.8 Gg and 56.0 Gg, respectively. The GEO LAI total emissions are more similar to the E1 results than those of the MODIS LAI. However, if only grid cells over which the MODIS LAI has no missing values are taken into account, the total BVOC emissions of E1-E3 are 63.5 Gg, 62.6 Gg, and 61.8 Gg; i.e., the MODIS MCD15A2 LAI and the GEO LAI lead to 1.4% and 2.6% difference with E1, respectively. The problem is that the E1 BVOC emissions in the region where the MODIS LAI has missing values account for 16.3% of the total E1 emissions. Considering the importance of BVOC emissions in suburban areas on air quality, the GEO and GLASS LAI may be better choices for use in BVOC estimation for regional air quality simulation and forecasting. In particular, the estimates obtained using the GEO LAI for specific BVOC species all differ from E1 by less than 4%.

The different PFT inputs used in experiments E4 and E5 lead to a 0.6% increase and a 26.3% decrease in total BVOC emissions, respectively. The extensive broadleaf tree cover in the MODIS MCD12Q1 LC dataset leads to higher BVOC emissions by way of a higher emission rate.

In this study, the cross-walking table used in the CCI-LC to convert LC classes to PFTs contains a scale factor that increases the proportion of grasses and decreases the proportions of other PFTs. This process leads to the total BVOC emissions in E4 being approximately 75% of the total E1 emissions. For E1, E4, and E5, the high spatial resolution (30 m) of FROM can compensate to some extent for the mixed pixel problem of the MODIS (500 m) and CCI-LC (300 m) medium-resolution sensor products in this study.

## 3.2 Temporal variations

Figure 6 shows the temporal variations in BVOC emission of the four groups for all experiments (E1 to E5). The temporal distribution is similar in all experiments. Summer and winter emissions account for 74.9%–76.9% and 0.26%–0.40%, respectively, of annual BVOC emissions. The differences between the inputs do not have a significant effect on the temporal variability of the BVOC emissions estimated by the MEGAN model. Moreover, the ratio of BVOC emissions between the summer and winter seasons is 185–295, compared with a ratio of 9.77 in the Pearl River Delta region (Wang et al., 2011) and 4.9 in Hong Kong (Leung et al., 2010). In temperate regions like Beijing, BVOC emissions display a very strong annual cycle, as there are almost no BVOC emissions in winter owing to the low winter biomass of temperate deciduous trees as well as low temperatures in winter. Additionally, the emissions differ more between experiments in summer than in winter because of the high emission amount in summer. The black lines represent the previously estimated emissions of isoprene and monoterpenes for 1998 from Wang et al. (2003). As shown in Figure 6, our results have similar temporal variability as the Wang et al. (2003) results, but all the results in this study are higher than their estimate. The ratios between our summer estimates and the results in Wang et al. (2003) for summer are 2.24–3.2 and 1.97–2.66 for isoprene and monoterpenes, respectively. There are multiple reasons for this large discrepancy between the two studies. Apart from differences in the inputs of the two studies, the significant development of forest and vegetation in the entire region during the last two decades may have played a significant role in the increase in BVOC emissions, leading to the higher estimates in this study. Ghirardo et al. (2016) used field surveys of tree numbers to estimate that BVOC emissions in the megacity region of Beijing doubled from 2005 to 2010. Furthermore, the increasing trend of BVOC emissions in Beijing is consistent with model estimates by Ren et al. (2017).

## 3.3 Spatial distribution

Since the contribution of summer BVOC emissions to the total annual emissions can reach about 75% and photochemistry is very active in summer, our analysis of spatial distributions is mainly focused on summer BVOC emissions.

Figure 7 displays the spatial distribution of the average emission rates of isoprene, monoterpenes, and sesquiterpenes during summer in E1 as well as the difference between E1 and the other experiments. According to (a), (f), and (k) in Figure 7, the BVOC emission hotspots are concentrated in the rural forest region around the city of Beijing. Despite the updating frequency of the GLASS datasets being 8 days and that of GEO v2 being 10 days, isoprene, monoterpenes, and sesquiterpenes of in E1 and E2 have

nearly identical spatial distributions. As mentioned in Sec. 2.2.2, the mask area of the MODIS LAI directly leads to missing BVOC emissions in the suburban area; consequently, the relatively low LAI values also lead to the slight decreases in isoprene, monoterpenes, and sesquiterpenes (Figure 7 (c), (h), and (m)).

The spatial distribution of BVOC emissions in E4 and E5 is conspicuously different than in E1, in keeping with differences in the PFT distributions in the inputs (Figure 3). E4 shows lower isoprene emissions than E1 in the northeast, the Huai Rou District, and Mi Yun County in Beijing, and higher isoprene emissions in the west, the Mengtou Gou District, and the Haidian District, which is likely due to the differences in the broadleaf tree distribution, the PFT with the highest emission potential, between the inputs for E1 and

E4. The E5 experiment, which used the CCI-LC, shows similar results to the E4 experiment with MODIS LC for isoprene, although the higher isoprene emissions in the west are more obvious for E5, reaching 1.2–1.8 mg m$^{-2}$ h$^{-1}$. E4 and E5 both estimate higher monoterpene and sesquiterpene emissions than E1 at the western edge of the boundary of Beijing and lower emissions in the northeast in Mi Yun County. Overall, the main BVOC emission pools are the regions to the west and northeast of the megacity, such

that the city is surrounded by the BVOC pools. In terms of $O_3$, although the forest area lacks NOx emissions, the oxides of isoprene, e.g., formaldehyde, could be transported to the city region and affect urban air quality (Geng et al., 2011); similarly, NOx from the urban area could be transported to the rural area and form $O_3$.

**3.4 Discussion**

**3.4.1 Sensitivity of BVOC emissions to LAI and PFT**

To study the effect of the LAI input on BVOC emissions, we adopted three independent satellite-derived LAI datasets. According to the direct validation by Xiao et al. (2016), the GLASS and the GEO LAI are generally of better quality than the MODIS MCD15A2 LAI. Although the average MODIS MCD15A LAI is lower than the GLASS and GEOv2 LAI, the comparison of BVOC emissions with E1 in the region

over which MODIS is valid (i.e., no missing values) showed that use of the GEOv2 and MODIS LAI input led to decreases of only 1.4% and 2.6%, respectively. The discrepancies between different LAI inputs do not obviously affect the estimate of BVOC emissions in Beijing. However, considering the missing values in the MODIS MCD15A2 LAI, using the GLASS LAI and the GEO LAI is a better solution than using interpolation to fill in the missing values in the MODIS LAI.

The discrepancies of PFT datasets are relative larger than that among the LAI datasets, therefore, the corresponding BVOC emission results showed more notable differences induced by PFT input. Which

LC dataset is used in the model significantly affects the BVOC emission estimates (Zhao et al., 2016;Wang et al., 2011). There are two major sources of uncertainty in the PFTs: the accuracy of the LC map and the cross-walking table used to convert the LC classes to PFTs (Hartley et al., 2017). 61 sample points in Beijing and the surrounding area from the Land Cover Validation Dataset by Zhao et al. (2014) were used as primary validation for the LC data sets. The validation samples were collected from TM/ETM images for 2009–2011. The accuracies of the FROM, MODIS, and CCI-LC datasets are 59.67%, 54.1%, and 50.81%, respectively. Since FROM LC has the same benchmark period as the validation dataset and similar spatial resolution, the FROM LC displayed better accuracy than the other two products. The validation results only can only coarsely assess the accuracy of the LC data sets. The advantage of the high-resolution data is that it diminishes the uncertainties associated with mixed pixels in the medium-resolution LC map, which relies on the cross-walking table to convert the LC classes to PFTs.

The uncertainties associated with the cross-walking table are more evident in CCI-LC. The cross-walking table used in E5 is the default table designed for the global scale. Therefore, two more sensitivity experiments were designed using the "minimum biomass" (minCW) and "maximum biomass" (maxCW) cross-walking tables provided by Hartley et al. (2017) for CCI-LC to examine the uncertainties associated with the cross-walking table. As shown in Table 6, the area fractions of broadleaf and needleleaf trees were 29.9% and 8.8% in the maxCW simulation, respectively, which are similar to those of FROM and MODIS (Figure S1), while the minCW simulation led to relatively low area fractions of 10.9% and 3.7% for broadleaf trees and needleleaf trees, respectively. The BVOC emission estimates with diverse cross-walking tables for CCI-LC are shown in Table 7. Compared with the results of E5, the maxCW and minCW simulations led to a 48.1% increase and a 44.7% decrease in isoprene and a 20.2% increase and a 33.3% decrease for monoterpene, respectively, indicating the strong effect of the cross-walking table on the BVOC estimates, which is more significant for the medium-resolution map. But for a high-resolution LC map based on TM/ETM images like FROM, high spatial resolution could diminish the uncertainty from cross walking processes. Furthermore, the BVOC emissions in the maxCW experiment are similar to the results of E1 with FROM LC and E4 with MODIS LC: a 7.0% increase for isoprene and 9.0% decrease for monoterpenes compared to E1, and a 1.0% increase for isoprene and 8.6% increase for monoterpenes compared to E4. Examining the features of local forests provided by high resolution LC map shows that the maxCW is likely a more accurate representation of the local PFTs. The BVOC emission estimates in the Beijing region using the three LC data sets are similar, falling within the ranges of 28.5–30.5 Gg for isoprene, 9.3–10.1 Gg for monoterpenes, 1.2–1.4 for sesquiterpenes, and 28.3–35.6 Gg for other VOCs.

### 3.4.2 Comparison with previous studies

Table 8 presents the BVOC emissions in Beijing estimated in this study and in previous studies. To facilitate comparison, the total emissions (Gg) are converted into the area average emission intensity $(g/m^2)$. As showed in Table 8, the isoprene results in this study are higher than the results in Wang et al. (2003), Klinger et al. (2002), and Ren et al. (2017) but lower than the results for 2010 in Ghirardo et al. (2016) and the calculation by Li and Xie (2014). All the monoterpene results, except for the results of Wang et al. (2003) and those of Ghirardo et al. (2016) for 2005, are in the range of 0.51–0.68.

The results of Klinger et al. (2002) as well as Li and Xie (2014) are subsets of the inventory for the whole China region, while the others are more thorough and concentrate specifically on the Beijing region. Ren et al. (2017) used the Global Biosphere Emissions and Interactions System (GLOBEIS) model framework (Guenther et al., 1999) to calculate the BVOC emissions, while Li and Xie (2014) and this study used the MEGAN model framework. The two models adopt the same algorithms to account for the environmental conditions (mainly based on Guenther et al. (1995)), but they treat the biomass and seasonal variance in different ways. The results of the MEGAN model are generally higher than the results of GLOBEIS model. Because of the local nature of this study and the effect of different resolutions, isoprene emissions in this study are slightly lower and monoterpene emissions are slightly higher than the results extracted from the national scale inventory by Li and Xie (2014). The sensitivity experiment using the default standard emission rates of the MEGAN model resulted in 80% higher isoprene emissions compared with E1, which shows that the regional results of Li and Xie (2014) may lead to some overestimation of the BVOC emissions.

The estimates made by Ren et al. (2017) used a species-level vegetation inventory based on field surveys, while this study used PFT-scale estimates based on satellite datasets, with statistically-derived PFT emission factors. The former method may be more accurate than the latter since emission factors differ between tree species, but it is limited by the rough process of data collection, and satellite based inventories are more easily gridded, facilitating coupling with chemistry transport models and thus allowing further investigation of the effect of BVOCs on atmospheric chemistry. Moreover, in Ren et al. (2017) the coverage of broadleaf trees and needleleaf trees is 18.7% and 7.4%, respectively, and in this study, it is 20.0%–30.0% and 6.3%–7.3%, respectively; i.e., the coverage of needleleaf trees, the main contributors to the emission of monoterpenes, is similar between the two studies, and thus the monoterpene emissions are relatively consistent. However, the isoprene estimates in this study are

generally higher than those of Ren et al. (2017): the ratios of isoprene emissions between the two studies is 1.15–1.7, which are similar to the ratios (1.06–1.6) of broadleaf tree area between the two studies. Furthermore, the diversity of meteorological inputs also contributes to the discrepancies among these studies. This study as well as Li and Xie (2014), both based on MEGAN model, adopted the mesoscale

model as meteorological inputs, and others used the in-situ data from stations. Considering the spatial diversities, e.g. terrain and landscape forcing, the meteorological model could be a reasonable approach to account for meteorological conditions of the whole region. But the simulation bias would correspondingly lead to bias of BVOC emission estimation. Wang et al. (2011) investigated the impact of meteorological simulation bias on estimating BVOC emissions by perturbing the simulated

temperature and radiation with their RMSE (T2/Radiation $\pm$ RMSE) for Pearl River Delta, China. And their sensitivity tests showed that the decreasing (increasing) temperature led to −19.2% (+26.7%) and −18.5% (+16.2%) difference of isoprene emission and monoterpene emissions, respectively; and decreasing (increasing) radiation led to −39.6% (+50.7%) and −14.3 (+16.8%) difference of isoprene emission and monoterpene emissions, respectively. These results indicate the simulation bias could affect

the BVOC emissions. But not all grids exist strong simulation bias, and impact of meteorological conditions would also be limited by other conditions, e.g. standard emission factor. The simulation bias among urban or suburban region, where the BVOC emissions are low, would not have an obvious effect on whole region emission estimation. But the evaluation results showed the overestimation of DSW in this study, which would correspondingly lead to the overestimation of light-dependent BVOC emissions.

In addition, Ghirardo et al. (2016) and Ren et al. (2017) reported that the increase of BVOC emissions because of local green land development. Ghirardo et al. (2016) showed that BVOC emissions doubled in the city region of Beijing from 2005 to 2010. Furthermore, Ren et al. (2017) found an even stronger increase in the urban region due to local green policy and favorable conditions. Ghirardo et al. (2016) and Ren et al. (2017) both investigated BVOC emissions from urban green space in the Beijing region.

Considering the strong anthropogenic emissions and anthropogenic forcings such as high temperatures and ozone pollution, BVOC emissions from urban green space may have a more direct and stronger impact on urban air quality than suburban and rural emissions. However, this is difficult to evaluate using a mesoscale model like MEGAN, which relies on satellite-based datasets. Therefore, the field-survey based research discussed above may play an important role in future studies concerning the impact of

urban BVOC emissions on air quality.

## 4. Conclusions and future work

The first step in investigating the effect of natural emissions on local air quality is to estimate a reliable BVOC emission inventory. In this work, we established an hourly, 3 km gridded inventory of BVOC emissions over Beijing in 2013 based on the latest MEGAN 2.1 model. The MEGAN model was driven by the WRF v3.3.1 model and several different satellite LAIs and PFTs were adopted to investigate and constrain the uncertainties of these input variables. Because the FROM LC product has the highest spatial resolution, the results of the experiment using the FROM LC and GLASS LAI datasets are treated as the baseline results.

(1) According to the results of the baseline experiment, the total quantity of BVOCs emitted in 2013 in Beijing was 75.9 Gg, with isoprene, monoterpenes, sesquiterpenes, and other VOCs accounting for 37.6%, 14.6%, 1.8% and 46.0% of the total, respectively. BVOC emissions in Beijing display strong temporal variability: the summer season contributes 74.9%–76.9% of the total emissions while the winter season only contributes 0.26–0.4% for all experiments. This is a result of the low temperatures and near-zero biomass of deciduous trees in winter.

(2) Different satellite LAI inputs were adopted to investigate the impact of LAI. MODIS MCD15A2 and GEO v2 LAI led to slight decreases in total BVOC emissions of 1.4% and 2.6%, respectively, over the region for which MODIS LAI is valid. The missing values in MODIS LAI led to an emission loss of 16.3% compared with E1, and compared to filling in the missing values by interpolation, the GEO and GLASS LAI products may be the better choice for regional BVOC emissions estimates. The differences between E1 and E2 for all BVOC species are lower than 4.0%, and spatial and temporal distributions are similar between the two experiments.

(3) The FROM-GLC, MODIS MCD12Q1 LC, and CCI-LC products were adopted to investigate the sensitivity of the model to LC data. Compared to E1, the results obtained using MODIS MCD12Q1 LC have similar total emissions and temporal variability but different spatial features. For CCI-LC, sensitivity tests using different cross-walking tables illustrated that the cross-walking table used to convert LC classes to PFTs has a clear impact on the BVOC emission estimates, with the total amount of BVOC emissions estimated ranging from 42.1 to 70.2 Gg. When using the "maximum biomass" cross-walking table, the CCI-LC PFTs are relatively consistent with other two PFT products, and the BVOC emissions in the Beijing region estimated using the three different LC data sets are similar, falling within the range of 28.5–30.5 Gg for isoprene, 9.3–10.1 Gg for monoterpenes, 1.2–1.4 for sesquiterpenes, and 28.3–35.6 Gg for other BVOCs.

(4) The BVOC emission estimates obtained in this study are much higher than earlier estimates (Wang et al., 2003;Klinger et al., 2002) and similar to those in a recent study by Ren et al. (2017). Ghirardo et al. (2016) and Ren et al. (2017) both reported the development of local green areas and the active greening policy could be an important driver stimulating increasing in BVOC emission. Furthermore, considering the changing of the BVOC emissions, the regional chemistry transport model will be used to investigate and evaluate the role of BVOC on local air pollution as the next step research.

**Data and codes availability**

The source code of WRF model V3.3.1 and MEGAN v2.1 is available at http://www2.mmm.ucar.edu/wrf/users/ and https://bai.ess.uci.edu, respectively. The FROM-GLC can be downloaded from the website of Department of Earth System Science, Tsinghua University, at http://data.ess.tsinghua.edu.cn/index.html. The CCI-LC can be downloaded from the website of Climate Change Initiative Program at https://www.esa-landcover-cci.org. The GLASS LAI can be obtained through the website of National Earth System Science Data Sharing Infrastructure at http://www.geodata.cn/thematicView/GLASS.html or the website of Global Land Cover Facility, University of Maryland, at http://glcf.umd.edu/data/lai/. The GEO v2 LAI is available on the website of the Copernicus Global Land Service at https://land.copernicus.eu/global/products/. The MODIS MCDQ12 LC and MODIS MCD15A2 LAI, Version 5, are available on the website of Land Process Distributed Active Center at https://lpdaac.usgs.gov/dataset_discovery/modis/modis_products_table.

**Acknowledgements**

The National Key R&D Program of China (2017YFC0209805), the National Natural Science Foundation of China (41305121), National Key Fundamental Research and Development Program of China (2014CB953903) and the Fundamental Research Funds for the Central Universities funded this work.

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

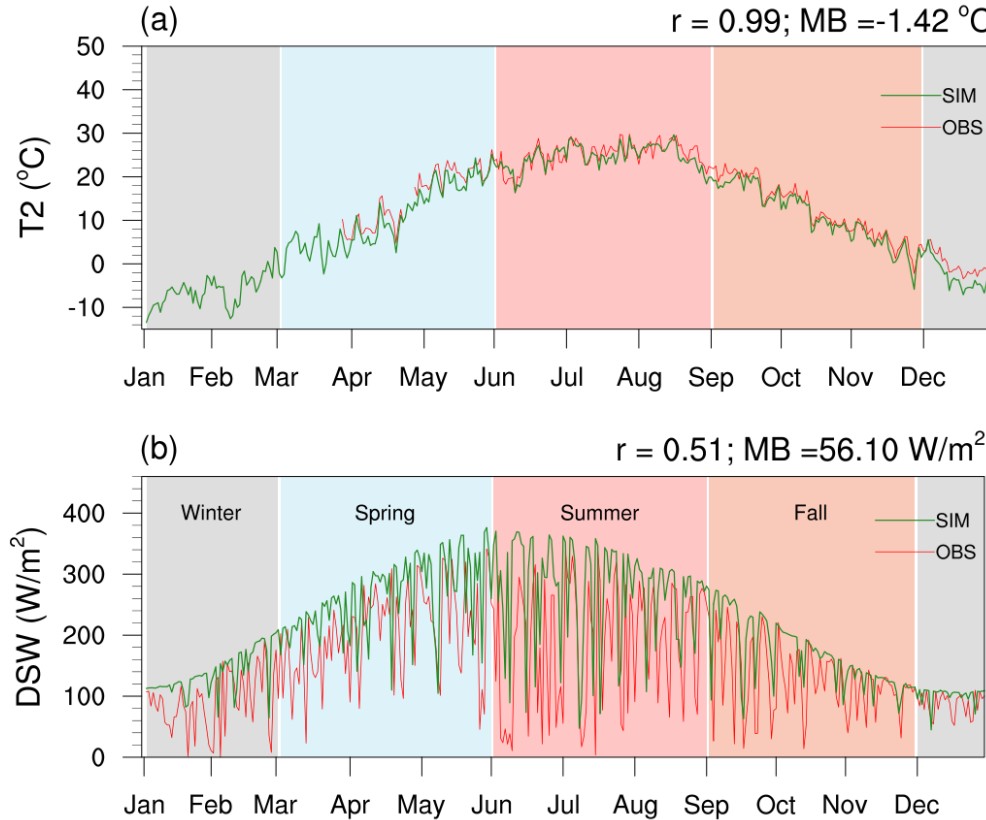

**Figure 1. Time-series plots of (a) station-averaged simulated/in-situ 2m temperature (T2) as well as (b) simulated/in-situ downward shortwave radiation (DSW) in Beijing Station. The r and ME represent correlation coefficient and mean bias, respectively.**

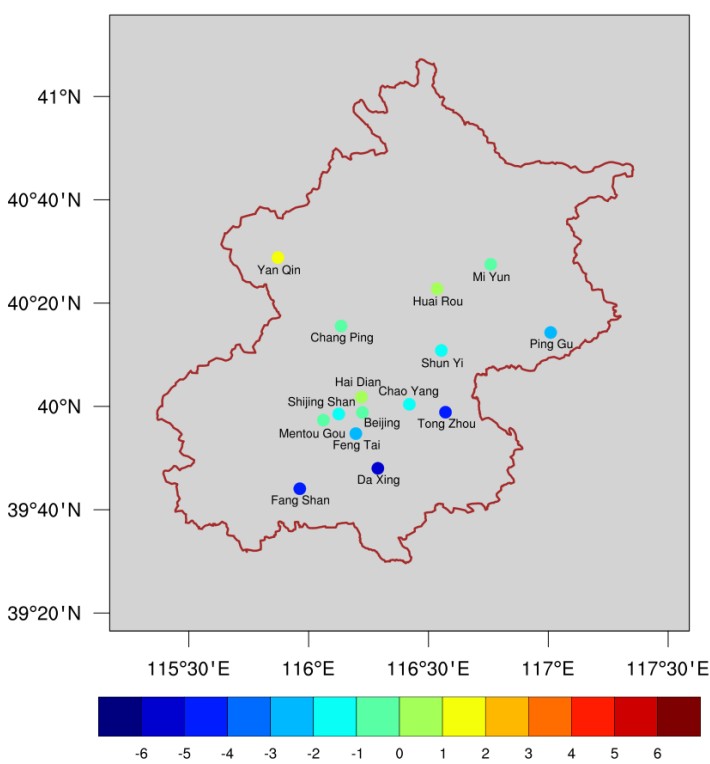

5    **Figure 2. Spatial distribution of the mean bias (MB) of temperature (T2) from all available sites.**

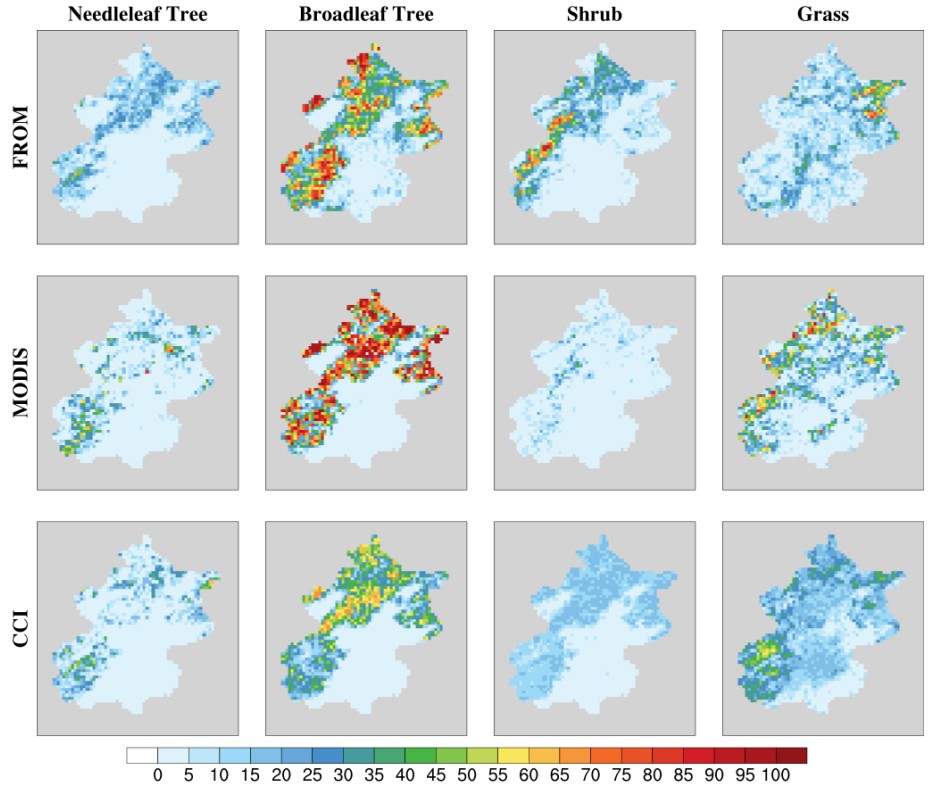

**Figure 3. Spatial distribution of the proportions of plant functional types (PFTs) in model grids in the three land cover inputs.**

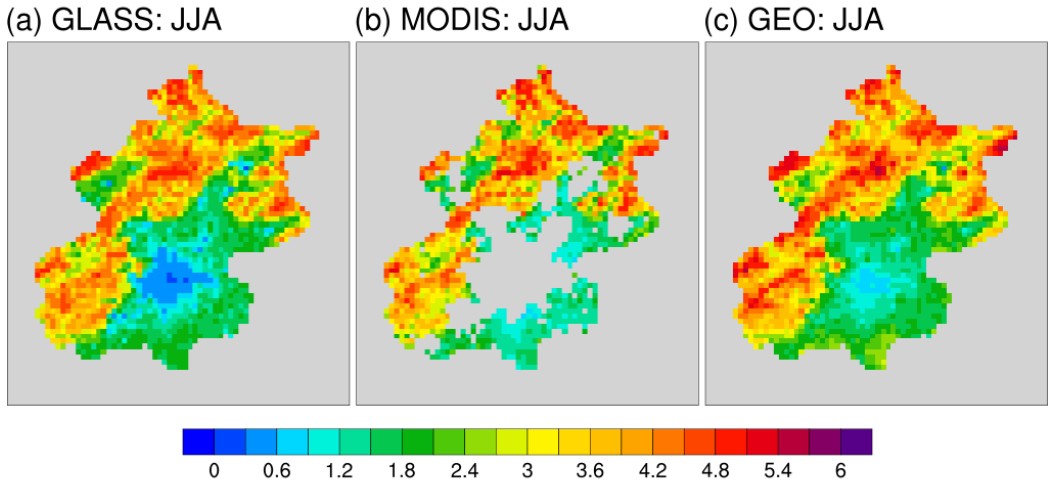

**Figure 4. The average spatial distribution of (a) GLASS, (b) MODIS, and (c) GEO v2 LAI in summer (June, July, and August; JJA).**

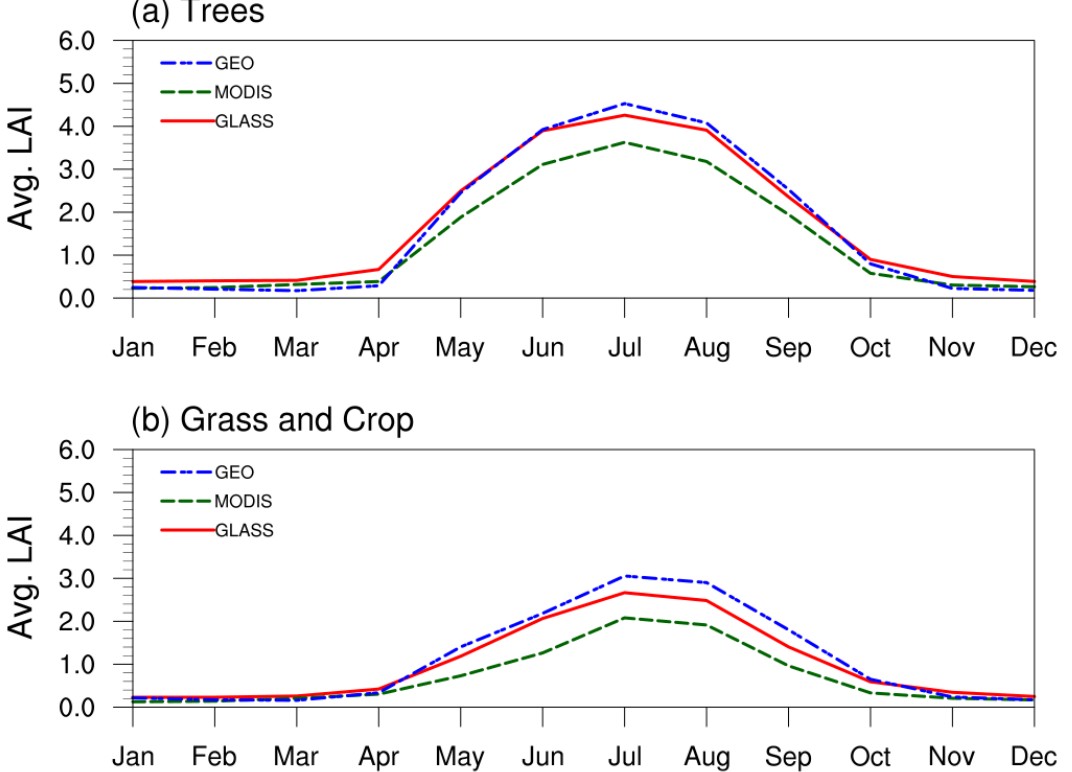

Figure 5. The monthly average leaf area index (LAI) values of (a) trees and (b) herbs in the three LAI products.

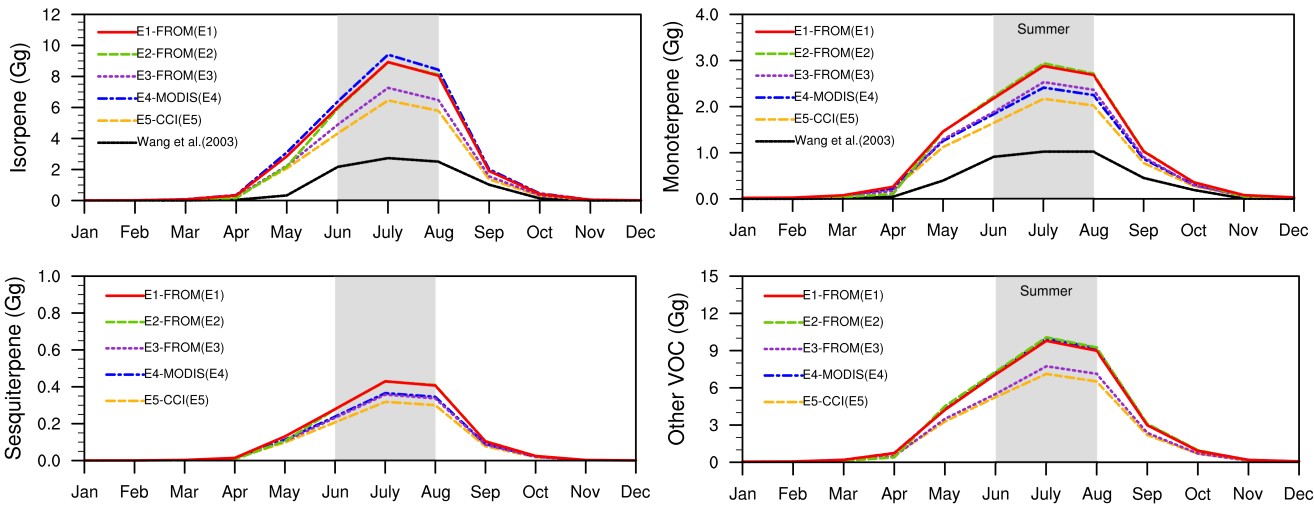

Figure 6. The temporal variability of biogenic volatile organic compounds (BVOCs) in all simulation experiments.

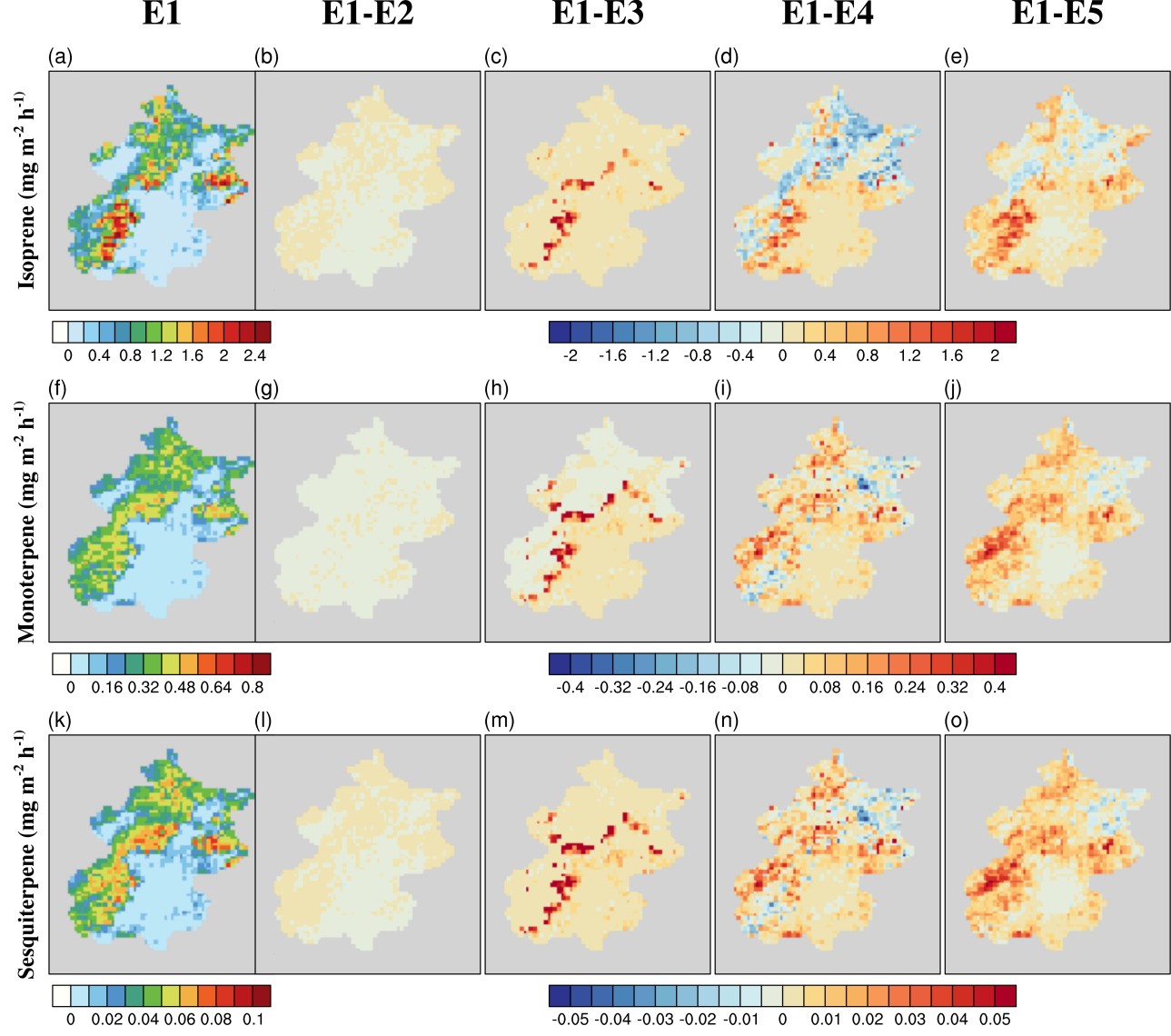

**Figure 7. The spatial distribution of average emissions of the three main VOC species in summer.**

**Table 1. Adjusted (original) isoprene emission factors (EF) for each plant functional type (PFT). (Unit: µg m-2 h-1)**

| PFT | Broadleaf Trees | Needleleaf Trees | Shrub | Grass | Other Crops | Corn |
|---|---|---|---|---|---|---|
| EF | 6510 (10000) | 64 (600) | 544 (4000) | 14 (800) | 2 (50) | 1 (1) |

**Table 2. Statistics of the primary verification of the temperature (T2) with in situ data. The ME, MB and RMSE are abbreviations for mean error, mean bias, and root mean square error, respectively.**

| Name | ME (°C) | MB (°C) | r | RMSE (°C) |
|---|---|---|---|---|
| Beijing | 1.97 | −0.2 | 0.97 | 2.48 |
| Hai Dian | 1.91 | 0.01 | 0.97 | 2.46 |
| Chao Yang | 2.27 | −1.15 | 0.96 | 2.82 |
| Shun Yi | 2.17 | −1.40 | 0.97 | 2.77 |
| Huai Rou | 1.98 | 0.19 | 0.97 | 2.51 |
| Tong Zhou | 4.86 | −4.84 | 0.97 | 5.44 |
| Chang Ping | 2.12 | −0.49 | 0.97 | 2.67 |
| Yan Qin | 2.84 | 1.81 | 0.96 | 3.46 |
| Feng Tai | 2.65 | −2.07 | 0.96 | 3.34 |
| Shijing Shan | 2.20 | −1.07 | 0.96 | 2.81 |
| Da Xing | 5.13 | −5.11 | 0.96 | 5.80 |
| Fang Shan | 4.56 | −0.50 | 0.96 | 5.34 |
| Mi Yun | 2.10 | −0.46 | 0.96 | 2.73 |
| Mengtou Gou | 2.06 | −0.56 | 0.96 | 2.64 |
| Ping Gu | 2.95 | −2.61 | 0.97 | 3.70 |

5    **Table 3. The area fractions of different plant functional types (PFTs) in Beijing from the three land cover datasets.**

| | Broadleaf Trees | Needleleaf Trees | Shrub | Grass | Other Crops | Corn |
|---|---|---|---|---|---|---|
| FROM-GLC | 27.3% | 7.3% | 11.3% | 11.9% | 3.6% | 23.7% |
| MODIS LC | 30.3% | 6.6% | 2.3% | 15.8% | 21.5% | 11.8% |
| CCI-LC | 20.0% | 6.3% | 8.4% | 16.6% | 7.6% | 5.6% |

**Table 4. The simulation experiment configurations. E1 is the baseline experiment, and E1-E3 were used to investigate the impact of leaf area index inputs. The impact of different plant functional type (PFT) inputs was investigated by E1, E4, and E5.**

|  | Land Cover | Leaf Area Index |
|---|---|---|
| E1 (Baseline) | FROM | GLASS v1.1 |
| E2 | FROM | GEO v2 |
| E3 | FROM | MODIS MCD15 |
| E4 | MODIS MCD12Q1 | GLASS v1.1 |
| E5 | CCI-LC | GLASS v1.1 |

**Table 5. The total annual BVOC emissions from all experiments. (Unit: Gg)**

|  | Isoprene | Monoterpenes | Sesquiterpenes | Other VOCs | SUM |
|---|---|---|---|---|---|
| E1 | 28.5 | 11.1 | 1.4 | 34.9 | 75.9 |
| E2 | 27.7 | 11.0 | 1.4 | 35.6 | 75.7 |
| E3 | 23.1 | 9.8 | 1.2 | 27.7 | 61.8 |
| E4 | 30.2 | 9.3 | 1.2 | 35.8 | 76.5 |
| E5 | 20.6 | 8.4 | 1.0 | 26.0 | 56.0 |

**Table 6. The area fractions of different plant functional types (PFTs) in Beijing from the CCI-LC with "Maximum Biomass", "Minimum Biomass" and default cross-walking tables.**

|  | Broadleaf Trees | Needleleaf Trees | Shrub | Grass | Other Crops | Corn |
|---|---|---|---|---|---|---|
| Max Biomass | 29.9% | 8.8% | 3.4% | 15.1% | 7.6% | 3.8% |
| Default(E5) | 20.0% | 6.3% | 8.4% | 16.6% | 7.6% | 5.6% |
| Min Biomass | 10.9% | 3.7% | 8.2% | 27.5% | 7.6% | 6.8% |

**Table 7. The estimations of BVOCs emission by adopting CCI LC with "Maximum Biomass", "Minimum Biomass" and default cross-walking tables. (Unit: Gg)**

|  | Isoprene | Monoterpenes | Sesquiterpenes | Other VOCs | SUM |
|---|---|---|---|---|---|
| Max Biomass | 30.5 | 10.1 | 1.3 | 28.3 | 70.2 |
| Default (E5) | 20.6 | 8.4 | 1.0 | 26.0 | 56.0 |
| Min Biomass | 11.4 | 5.6 | 0.7 | 24.4 | 42.1 |

**Table 8. Comparison of the average emission intensities of isoprene and monoterpenes from this study and previous publications.**

| | Year | Area (km$^2$) | Isoprene (g m$^{-2}$) | Monoterpenes (g m$^{-2}$) |
|---|---|---|---|---|
| Wang et al. (2003) | 1998 | 16400 | 0.54 | 0.24 |
| Klinger et al. (2002) | 2002 | 16400 | 0.96 | 0.54 |
| Li & Xie (2014) | 2003 | 16400 | 1.91 | 0.50 |
| Ghirardo et al. (2016) | 2005 (city level) | 1434 | 2.05 | 0.28 |
| Ghirardo et al. (2016) | 2010 (city level) | 1434 | 4.14 | 0.63 |
| Ren et al. (2017) | 2015 | 16400 | 1.08 | 0.66 |
| This Study | 2013 | 16400 | 1.25–1.84 | 0.51–0.68 |