# Peer review of "Sensitivity of Biogenic Volatile Organic Compound Emissions to Leaf Area Index and Land Cover in Beijing"

_Atmospheric Chemistry and Physics, 2017_

## Referee Comment (RC1) · Anonymous Referee #2 · 21 Dec 2017

**Review of "Sensitivity of Biogenic Volatile Organic Compounds Emissions to Leaf Area Index and Land Cover in Beijing" by Wang et al., 2017**

The authors report the sensitivity of WRF (v3.3.1)-MEGAN (v2.1) calculated BVOCs emissions to land cover and LAI inputs over the Beijing area in 2013. The results were compared with previous studies and related to regional air quality. This material is original and suitable for ACP.

The methodology section needs to be significantly expanded to include more descriptions on experiment setup (see my specific comments below). Any novel settings should be highlighted.

The paper can benefit from careful language editing, preferably with help from a native English speaker. I recognize that the authors made some efforts to address this issue brought up during the ACPD quick report phase. However, the current version still contains many grammar mistakes and awkward sentences. References are often inaccurate/inappropriate. Transitions from one sentence to another, from one paragraph to another are not smooth. Also, when one paragraph ends and a new one begins, the authors should either indent the first line of the new paragraph, or leave a line space between the two paragraphs. Here are some suggested edits to the first sentences of your abstract.
P1, L11: air quality pollution → air pollution
P1, L12: delete "still", and also requires other emission inventories.. A sentence saying BVOC emissions are sensitive to land and met conditions should be placed here.
P1, L15-16: "based on" → "using"; add "the" before "Model of ….."; delete "model" after v2.1
P1, L19: "are used to design five experiments, as E1-E5, to calculate and test the sensitivity of the model" → are used in five model sensitivity experiments, as E1-E5
P1, L20: "Based on the meteorological conditions from Weather Forecasting and Research (WRF) model, this inventory is an hourly inventory with 3-km spatial resolution."→These sensitivity calculations were driven by hourly, 3 km meteorological fields from the Weather Forecasting and Research (WRF) model.

A number of specific comments and suggestions are given below:

1) Novelty: The authors argue that using spatially and temporally varying meteorological fields output from WRF is advantageous, compared with the approaches in some previous studies. Using WRF fields to drive MEGAN calculations is not at all a novel approach and has been widely used in a large number of studies, including some cited by the authors. As the authors are already aware, the uncertainty in their WRF simulation contributed to the estimated BVOC emission biases. There is no need to emphasize the benefit of driving MEGAN using WRF. Rather, if any novel configurations were applied to your WRF simulation, which helped reduce errors in the modeled T2, radiation, moisture, etc, they should be highlighted. See also my next comment.

2) More information regarding your WRF simulation and evaluation approach should be provided in Section 2.2.1. These should include:
- introduce the initialization time for each domain
- introduce the vertical spacing for each domain

- introduce your physics options for each domain and their suitability for the Beijing area (based on literature and/or any sensitivity simulations the authors may have conducted)
- introduce the land cover and vegetation dynamics (e.g., green vegetation fraction) input data, including the year(s) these input data represented, and discuss how they may have contributed to biases from your WRF simulation.
- P4, L23: Skamarock et al., 2005 is for WRF version 2. Please cite WRF version 3 documentation.
- P4, L27: justify "we considered the second day as the reasonable results", for example, compare the 1-day and 2-day model performance.
- P4, L29: explain why daily T2 was evaluated, instead of hourly T2? Change "among" to "within"
- P5, L1/L4: unit is missing for these biases. Why was the MB of -1.5 degree mentioned twice?
- P5, L9: which single station?

3) Issues regarding satellite products:
- P6, L22: "Because of the highest spatial resolution of the FROM LC product, the experiment using FROM PFT and GLASS LAI as inputs is the baseline experiment (E1)". I don't understand the logical connections between resolution and choice of the baseline experiment.

- Although the land cover datasets used in this study differed by at least a factor of ten in resolution (30m vs 500/300m), they are all at much finer resolution than the 3 km WRF-MEGAN grid. It would be helpful to explain how these data were regridded to your WRF-MEGAN model grid. This would help us understand how the original data resolution may have affected your results. Approach used to reproject the original 1 km LAI data should also be provided. Missing a "respectively" in P5, L25.

- P5, Section 2.2.2: The land cover and LAI data citations are not helpful. For each dataset, please cite the corresponding algorithm/validation paper, and provide the dataset doi or/and accurate links to retrieve the data. MOD15 is not an accurate description for the used MODIS LAI data.  I assume the correct format should be MCD15A2H, Collection/Version XX. Same issue exsits in Table 4.

- Did the author screen the LAI data and if so, based on what criteria? What values were used for grids with missing data/unrealistic (e.g., extremely high) LAI? Previously studies have reported MEGAN sensitivity to PFT and LAI, so it'd be helpful to compare your findings with theirs.

4) Section 2.1: Some introductions on how LAI impacts the MEGAN calculations should be added. Using equations consistent with Guenther et al. (2012) and Sindelarova et al. (2014) is recommended. Equations should be numbered and referenced in text. Please also clarify how γsm and ρ were handled in this study.

5) The uncertainty section (3.5) is not well written, and the current discussions are very qualitative and not informative.

6) P2, L25-27: Please provide the source for "the statistical data from the Nation Forest Resources Survey (NFRS) reported that the forest coverage rate in Beijing rose from 20.6% to

35.8% during 1998-2013". This is also the right place to mention the impact of different met conditions during the earlier periods and 2013.

7) P1, L14; P2, L28; P3, L1: "new" is not accurate. Previous estimates of BVOCs emissions introduced by the authors are not for 2013.

8) P5, L25: Shouldn't the last sentence belong to Section 2.2.1? Define MCIP, and use the correct link for MCIP.

8) P5, L28: four species → four groups

9) P9, Section 3.3: Method of this sensitivity test should be first introduced in Section 2.

10) To comply with the ACP policy, data availability should be included in the "Acknowledgements" section.

11) Captions of Figures 4, 6, 7: specify which experiment these were based on.

---

## Referee Comment (RC2) · Anonymous Referee #1 · 7 Jan 2018

This manuscript describes a modeling study of biogenic VOC (BVOC) emissions from Beijing China. Since BVOC emissions are important for determining atmospheric composition and chemistry and are not well understood, this original study has the potential to contribute to the scientific understanding on this significant topic. The manuscript is difficult to understand in many places but that could be addressed with a thorough language editing.

The authors apply the MEGAN model, driven with WRF meteorology as is typically the case for MEGAN simulations. The most valuable part of the study is the investigation of the two of the main drivers of BVOC emissions: meteorology and landcover. For

meteorology, the authors compare temperature to observations and report a bias of cool temperature simulated by the model that is likely because the model does not adequately simulate the impact of the "urban heat island" on temperature. This is one of the more interesting results of the study and is a topic that the authors could potentially explore further with a more detailed examination and discussion of the canopy and leaf temperature simulations. For landcover, the authors compare different satellite based datasets. They do not compare with any in-situ observations so it is a sensitivity study with limited insights regarding accuracy and uncertainties.

The six main conclusions of the paper are listed in the conclusions section:

Conclusions #1 to 3 and #6 relate to the total emissions and the contribution of individual seasons. This would be of more interest if the study included some comparisons to BVOC emission measurements, so we have some idea if the emission results are correct. Since the paper does not include any observations of BVOC emissions, the MEGAN predictions of Beijing emissions should not themselves be the major focus of the manuscript. The current manuscript text (in the conclusion and elsewhere) devoted to describing the MEGAN model results (totals, seasonal and spatial variations) is too long and should be provided only as a brief description in the manuscript, and could perhaps be included in more detail in a supplemental section.

Conclusions #4 (LAI) and 5 (PFT) are the potentially more interesting contributions. However, there are several issues regarding the results and associated conclusions.

Page 11, line 24/25 states that MODIS LAI led to a 17.4% decline of total BVOCs compared with baseline in this study, because of the relatively big mask area in the MODIS LAI product. This is not a reasonable comparison. The mask indicates that no data is being provided for the masked region so it doesn't make sense to compare them. The default MEGAN LAI data on the MEGAN website replaces the MODIS LAI in the masked region with values based on an interpolation from the surrounding region. You could use this or some other approach but it is misleading to indicate that the

[Figure]

MODIS LAI is lower as indicated in the conclusion section and elsewhere (e.g., Figure 3).

Page 11, line 27/28 The statement, "Generally, the uncertainty of LAI is limited under the MEGAN model frame", is unclear but seems to suggest that because the GEO and GLASS LAI data products are similar that means that LAI uncertainties do not contribute substantially to MEGAN BVOC emission uncertainties. This is not necessarily the case as it probably just shows that the two datasets are based on a similar approach (with similar errors).

Regarding conclusion #5, and the PFT comparison in general, the authors apparently consider only the relative contribution of PFTs to the vegetation covered regions and do not consider the differences in total vegetation cover. I assume this is the case since the PFTs in table 3 add up to 100% but I expect the vegetation cover in Beijing must be less than 100%. How does total vegetation cover differ between the three landcover databases? In addition, the conclusion #5 reports the PFT cover differences but does not provide any insights on which is the most accurate, how uncertain they are, and what the implications are for modeling. For example, how important is it to get the relative PFT correct in comparison to getting total vegetation cover correct or accounting for the variability of emission factors within each PFT (i.e., not all broadleaf trees have the same isoprene emission factor).

Finally, it is evident that the modeling exercise described in this manuscript generally supports the results and conclusions of a similar study by Ren et al. (http://dx.doi.org/10.1016/j.envpol.2017.06.049) for the same region (Beijing) that covers the same topic more thoroughly. The Ren et al. paper is not referenced in this manuscript which is not surprising since it was only recently published. However, it is important that the authors do compare with and discuss the results and conclusions of the Ren et al. paper and consider whether (and how) their manuscript adds any new information to the existing scientific literature.

---

## Author Comment (AC1) · 24 Mar 2018

The authors thank the reviewer for his precious time and the constructive comments. Our detailed responses to the editor and referee comments are given below.

**General comment**

1) This manuscript describes a modeling study of biogenic VOC (BVOC) emissions from Beijing China. Since BVOC emissions are important for determining atmospheric composition and chemistry and are not well understood, this original study has the potential to contribute to the scientific understanding on this significant topic. The manuscript is difficult to understand in many places but that could be addressed with a thorough language editing.

Response: The authors appreciate your precious time and comments. As mentioned in the manuscript, this study is to investigate and discuss the sensitivities of MEGAN model using multiple satellite-based datasets. Therefore, it is necessary to evaluate the effect from input data on the estimation of BVOC emissions. **And the professional language editing has been called before the revised manuscript submitted to solve the language issue.**

[Figure]

**Language Editing Services**

*Registered Office:*
Elsevier Ltd
The Boulevard, Langford Lane,
Kidlington, OX5 1GB, UK.
Registration No. 331566771

**To whom it may concern**

The paper "Sensitivity of Biogenic Volatile Organic Compounds Emissions to Leaf Area Index and Land Cover in Beijing" by Hui Wang, Qizhong Wu was edited by Elsevier Language Editing Services.

Kind regards,

**Elsevier Webshop Support**

2) The authors apply the MEGAN model, driven with WRF meteorology as is typically the case for MEGAN simulations. The most valuable part of the study is the investigation of the two of the main drivers of BVOC emissions: meteorology and landcover. For meteorology, the authors compare temperature to observations and report a bias of cool temperature simulated by the model that is likely because the model does not adequately simulate the impact of the "urban heat island" on temperature. This is one of the more interesting results of the study and is a topic that the authors could potentially explore further with a more detailed examination and discussion of the canopy and leaf temperature simulations. For land cover, the authors compare different satellite based datasets. They do not compare with any in-situ observations so it is a sensitivity study with limited insights regarding accuracy and uncertainties.

Response: The authors appreciate the reviewer's comments. Regarding with meteorology conditions, in the manuscript, we did some validation to indicate the reasonability of simulation to estimate the BVOCs emission. The reviewer mentioned that exploring the impact of Urban Heat Island (UHI) may be an interesting topic, however, the most of available satellite-based land cover datasets, like MODIS 12Q1, can't present the green land among the city region for the limitation of spatial resolution. In this study, only the Finer Resolution Observation and Monitoring

of Global Land Cover (FROM-GLC) with 30m resolution could primarily characterize the major green land in the city, but the scattered green space like road greening can't be recognized, which make it not wise enough to discover this topic. The field surveys can provide the more thorough data of urban green space and the multiple studies have adopted such method to discover the relevant topic (Ren et al., 2017;Ghirardo et al., 2016;Chang et al., 2012), and this study focus on the typical landscape, which could be distinguished in the satellite land cover products.

The validations of the land cover datasets were done by the land-cover validation dataset from Zhao et al. (2014), and the baseline period of validation dataset is 2009-2011. The 61 available sample points from the above dataset in Beijing and its surrounding region were used and the results are showed in Table 1. The validation showed that the FROM-GLC has the highest accuracy of 59.67% among these land cover datasets. Since FROM LC has the same benchmark period with validation dataset and close spatial resolution, the FROM LC showed the better accuracy than the other two products. Considering effect of the spatial resolution and benchmark time, the validation results only indicate the reasonability of the land covers coarsely.

Table 1. Accuracies calculated based on the sample points from land-cover validation dataset.

|  | FROM-GLC | MODIS 12Q1 2013 | CCI LC 2013 |
|---|---|---|---|
| Accuracy | 59.67% | 54.10% | 50.81% |

**Specific suggestions and comments:**

1) Conclusions #1 to 3 and #6 relate to the total emissions and the contribution of individual seasons. This would be of more interest if the study included some comparisons to BVOC emission measurements, so we have some idea if the emission results are correct. Since the paper does not include any observations of BVOC emissions, the MEGAN predictions of Beijing emissions should not themselves be the major focus of the manuscript. The current manuscript text (in the conclusion and elsewhere) devoted to describing the MEGAN model results (totals, seasonal and spatial variations) is too long and should be provided only as a brief description in the manuscript, and could perhaps be included in more detail in a supplemental section.

Response: Thanks for your comments. We noticed that BVOCs observations that can be found in previous publication (Shao et al., 2009;Xie et al., 2008;Wu et al., 2016;Li et al., 2015) are mainly for the sites in Peking University (PKU) and Yufu, a city site and a rural site. These publications only provided the average value and standard deviation, but are not for year 2013. To evaluate the BVOCs emission and effect, the chemistry transport model (CTM) should be used (Geng et al., 2011;Zhao et al., 2016), but there is no time-serious observation of BVOCs in 2013, and the observation sites are not located in the forest region. According to the reviewer's comments, and partly decreased the descriptions of model results and added more details of the discussion of the sensitivity tests. Meanwhile, we adjusted the standard emission factor based on Ren et al. (2017) to discuss and compare the results of two similar studies.

2) Conclusions #4 (LAI) and 5 (PFT) are the potentially more interesting contributions. However, there are several issues regarding the results and associated conclusions. Page 11, line 24/25 states that MODIS LAI led to a 17.4% decline of total BVOCs compared with baseline in this study, because of the relatively big mask area in the MODIS LAI product. This is not a reasonable comparison. The mask indicates that no data is being provided for the masked region so it doesn't make sense to compare them. The default MEGAN LAI data on the MEGAN website replaces the MODIS LAI in the masked region with values based on an interpolation from the surrounding region. You could use this or some other approach but it is misleading to indicate that the MODIS LAI is lower as indicated in the conclusion section and elsewhere (e.g., Figure3).

Response: Thanks for your comments. We have further compared the effect of LAI products by considering two aspects, masking area and LAI value discrepancy. Firstly, we compared region that is available in MODIS LAI products, and it could explain the effect from the discrepancy of the LAI value on the BVOCs emission estimation. In addition, according to the Xiao et al. (2016), the direct validation with in-situ observation shows that the GLASS LAI has most similar results with the site observation, and the MODIS LAI is the worst. Secondly, the BVOCs discrepancy from the masking area of MODIS LAI is isolated. Since the MODIS adopting the vegetation canopy radiation model to produce LAI products (Knyazikhin et al., 1999), the region assigned as non-pure vegetation type would be treated as a missing value. Meanwhile, in this study, we used L4 level satellite products, and the producer of datasets has finished the work of interpolating the missing value in a reasonable range. Therefore, adopting the method like interpolating for the missing value in MODIS LAI is not helpful to improve the quality of the datasets but lead into new source of error, and we separately discussed the discrepancy within MODIS and other LAI products.

3) Page 11, line 27/28 The statement, "Generally, the uncertainty of LAI is limited under the MEGAN model frame", is unclear but seems to suggest that because the GEO and GLASS LAI data products are similar that means that LAI uncertainties do not contribute substantially to MEGAN BVOC emission uncertainties. This is not necessarily the case as it probably just shows that the two datasets are based on a similar approach (with similar errors).

Response: Thanks for your comments. The authors have followed the reviewer's comments. And this statement has been deleted from the manuscript.

4) Regarding conclusion #5, and the PFT comparison in general, the authors apparently consider only the relative contribution of PFTs to the vegetation covered regions and do not consider the differences in total vegetation cover. I assume this is the case since the PFTs in table 3 add up to 100% but I expect the vegetation cover in Beijing must be less than 100%. How does total vegetation cover differ between the three landcover databases? In addition, the conclusion #5 reports the PFT cover differences but does not provide any insights on which is the most accurate, how uncertain they are, and what the implications are for modeling. For example, how important is it to get the relative PFT correct in comparison to getting total vegetation cover correct or accounting for the variability of emission factors within each PFT (i.e., not all broadleaf trees have the same isoprene emission factor).

Response: Thanks for your comments. The data in table 3 has been corrected to the fractions of area of the different land covers to the total area of the Beijing region. The vegetation distribution is the key determinant of the standard emission factor. Satellite-based land cover products could provide the gridded spatial distribution of major landscapes, but it is limited to provide the further detail of the species information of different vegetation. Therefore, this approach is not suitable to solve the problem, which is mentioned by the reviewer that different species with same PFT have diverse emission factors. But the results based on satellite-based land cover map are gridded and available for the chemistry model directly, and such method is also the most common way for the researches of adopting CTM to investigate the topics about the air quality (Gao et al., 2016;Situ et al., 2013;Wei et al., 2018).

On the other hand, the field surveys can provide more information of species compositions, but the accurate spatial distribution of species is not available. The work done by Ren et al. (2017), mentioned by the reviewer, adopted the statistic species data from field surveys at administrative-region scale. And this approach may be more thorough to estimate the total BVOCs emission, but how to gridded these results and make it suitable for the CTM is not presented in his manuscript.

In general, adopting the satellite-based PFT map would lead to the errors from the species diversity of emission factor, but it is easier to be gridded for the following research. A compromising way is to estimate the emission factor of PFTs based on the statistical data of vegetation species, which presents the average emission factor of the

PFTs. And this method was also adopted by Wang et al. (2011) to provide reasonable parameters for the estimation of regional BVOCs emission.

In addition, the classification of the satellite land cover is also play a key role in determining the emission factor. The PFTs scheme in MEGAN v2.1 are from Community Land Model v4 (CLM4)(Lawrence et al., 2011), and it is significant to convert the diverse land cover classes to the PFTs, which is called cross-walking. The MODIS MCD12Q1 LC and the Climate Change Initiative Land Cover (CCI-LC) adopt the corresponding cross-walking tables to convert its classification system to PFTs. The FROM-GLC product doesn't provide the corresponding cross-walking table, therefore, the converting of FROM-GLC is based on its original classification system. Since pixels of the medium-resolution land cover datasets would contain the information of multiple land cover types and that the cross-walking table is one of the sources of uncertainty in land surface model (Hartley et al., 2017), we treated the 30m resolution FROM-GLC as the baseline land cover in the experiments, and adopted the other two land cover products to discover the impact of the discrepancy of land cover on the estimation of BVOCs emission. And in addition, we also used the CCI-LC to do the sensitivity tests of cross-walking table, and the results will be added to the Discussion section in the revised manuscript and the supplement.

Finally, it is evident that the modeling exercise described in this manuscript generally supports the results and conclusions of a similar study by Ren et al. (http://dx.doi.org/10.1016/j.envpol.2017.06.049) for the same region (Beijing) that covers the same topic more thoroughly. The Ren et al. paper is not referenced in this manuscript which is not surprising since it was only recently published. However, it is important that the authors do compare with and discuss the results and conclusions of the Ren et al. paper and consider whether (and how) their manuscript adds any new information to the existing scientific literature.

Response: The authors thanks for the reviewer's comments. Ren et al. (2017) presented the similar research about the BVOCs emission in Beijing during 2015. The two studies adopting the similar algorithms but different data sources. As mentioned above, Ren et al. (2017) adopted the statistical data of the main tree species and collected thorough parameter of the vegetation, which contains more detail compared with our previous data. Therefore, we adjusted our standard emission factor based on the data from Ren et al. (2017) and recalculated the BVOCs emission. The corresponding results and analysis would be presented in the revised paper. As emphasized by the reviewer, the results of Ren et al. (2017) is more thorough, and the comparison of two studies could be helpful to understand the discrepancy of estimations of MEGAN model and more accurate species-based estimation.

**Reference**

Chang, J., Ren, Y., Shi, Y., Zhu, Y., Ge, Y., Hong, S., Jiao, L., Lin, F., Peng, C., Mochizuki, T., Tani, A., Mu, Y., and Fu, C.: An inventory of biogenic volatile organic compounds for a subtropical urban–rural complex, Atmospheric Environment, 56, 115-123, https://doi.org/10.1016/j.atmosenv.2012.03.053, 2012.

Gao, M., Carmichael, G. R., Wang, Y., Saide, P. E., Yu, M., Xin, J., Liu, Z., and Wang, Z.: Modeling study of the 2010 regional haze event in the North China Plain, Atmos. Chem. Phys., 16, 1673-1691, 10.5194/acp-16-1673-2016, 2016.

Geng, F., Tie, X., Guenther, A., Li, G., Cao, J., and Harley, P.: Effect of isoprene emissions from major forests on ozone formation in the city of Shanghai, China, Atmospheric Chemistry and Physics, 11, 10449-10459, 10.5194/acp-11-10449-2011, 2011.

Ghirardo, A., Xie, J., Zheng, X., Wang, Y., Grote, R., Block, K., Wildt, J., Mentel, T., Kiendler-Scharr, A., Hallquist, M., Butterbach-Bahl, K., and Schnitzler, J.-P.: Urban stress-induced biogenic VOC emissions and SOA-forming potentials in Beijing, Atmospheric Chemistry and Physics, 16, 2901-2920, 10.5194/acp-16-2901-2016, 2016.

Hartley, A. J., MacBean, N., Georgievski, G., and Bontemps, S.: Uncertainty in plant functional type distributions

and its impact on land surface models, Remote Sensing of Environment, 203, 71-89, https://doi.org/10.1016/j.rse.2017.07.037, 2017.

Knyazikhin, Y., Glassy, J., Privette, J. L., Tian, Y., Lotsch, A., Zhang, Y., Wang, Y., Morisette, J. T., P.Votava, Myneni, R. B., Nemani, R. R., and Running, S. W.: MODIS Leaf Area Index (LAI) and Fraction of Photosynthetically Active Radiation Absorbed by Vegetation (FPAR) Product (MOD15) Algorithm Theoretical Basis Document, 1999.

Lawrence, D. M., Oleson, K. W., Flanner, M. G., Thornton, P. E., Swenson, S. C., Lawrence, P. J., Zeng, X., Yang, Z. L., Levis, S., and Sakaguchi, K.: Parameterization improvements and functional and structural advances in Version 4 of the Community Land Model, Journal of Advances in Modeling Earth Systems, 3, 365-375, 2011.

Li, J., Xie, S. D., Zeng, L. M., Li, L. Y., Li, Y. Q., and Wu, R. R.: Characterization of ambient volatile organic compounds and their sources in Beijing, before, during, and after Asia-Pacific Economic Cooperation China 2014, Atmos. Chem. Phys., 15, 7945-7959, 10.5194/acp-15-7945-2015, 2015.

Ren, Y., Qu, Z., Du, Y., Xu, R., Ma, D., Yang, G., Shi, Y., Fan, X., Tani, A., Guo, P., Ge, Y., and Chang, J.: Air quality and health effects of biogenic volatile organic compounds emissions from urban green spaces and the mitigation strategies, Environmental Pollution, 230, 849-861, https://doi.org/10.1016/j.envpol.2017.06.049, 2017.

Shao, M., Lu, S., Liu, Y., Xie, X., Chang, C., Huang, S., and Chen, Z.: Volatile organic compounds measured in summer in Beijing and their role in ground‐level ozone formation, Journal of Geophysical Research: Atmospheres, 114, 2009.

Situ, S., Guenther, A., Wang, X., Jiang, X., Turnipseed, A., Wu, Z., and Bai, J.: Impacts of seasonal and regional variability in biogenic VOC emissions on surface ozone in the Pearl River delta region, China, Atmospheric Chemistry and Physics, 13, 11803-11817, 2013.

Wang, X., Situ, S., Guenther, A., Chen, F. E. I., Wu, Z., Xia, B., and Wang, T.: Spatiotemporal variability of biogenic terpenoid emissions in Pearl River Delta, China, with high-resolution land-cover and meteorological data, Tellus B, 63, 241-254, 10.1111/j.1600-0889.2010.00523.x, 2011.

Wei, W., Lv, Z. F., Li, Y., Wang, L. T., Cheng, S., and Liu, H.: A WRF-Chem model study of the impact of VOCs emission of a huge petro-chemical industrial zone on the summertime ozone in Beijing, China, Atmospheric Environment, 175, 44-53, https://doi.org/10.1016/j.atmosenv.2017.11.058, 2018.

Wu, R., Li, J., Hao, Y., Li, Y., Zeng, L., and Xie, S.: Evolution process and sources of ambient volatile organic compounds during a severe haze event in Beijing, China, Science of The Total Environment, 560-561, 62-72, https://doi.org/10.1016/j.scitotenv.2016.04.030, 2016.

Xiao, Z., Liang, S., Wang, J., Xiang, Y., Zhao, X., and Song, J.: Long-Time-Series Global Land Surface Satellite Leaf Area Index Product Derived From MODIS and AVHRR Surface Reflectance, IEEE Transactions on Geoscience and Remote Sensing, 54, 5301-5318, 10.1109/tgrs.2016.2560522, 2016.

Xie, X., Shao, M., Liu, Y., Lu, S., Chang, C.-C., and Chen, Z.-M.: Estimate of initial isoprene contribution to ozone formation potential in Beijing, China, Atmospheric Environment, 42, 6000-6010, 2008.

Zhao, C., Huang, M., Fast, J. D., Berg, L. K., Qian, Y., Guenther, A., Gu, D., Shrivastava, M., Liu, Y., and Walters, S.: Sensitivity of biogenic volatile organic compounds to land surface parameterizations and vegetation distributions in California, Geoscientific Model Development, 9, 1959-1976, 2016.

Zhao, Y., Gong, P., Yu, L., Hu, L., Li, X., Li, C., Zhang, H., Zheng, Y., Wang, J., Zhao, Y., Cheng, Q., Liu, C., Liu, S., and Wang, X.: Towards a common validation sample set for global land-cover mapping, International Journal of Remote Sensing, 35, 4795-4814, 10.1080/01431161.2014.930202, 2014.

---

## Author Comment (AC2)

The authors thank the reviewer for his precious time and the constructive comments. The detailed responses to the editor and referee comments are given below.

**General comment**

1) The authors report the sensitivity of WRF (v3.3.1)-MEGAN (v2.1) calculated BVOCs emissions to land cover and LAI inputs over the Beijing area in 2013. The results were compared with previous studies and related to regional air quality. This material is original and suitable for ACP.

Response: The authors appreciate your precious time and effort to improve the quality of our manuscript. The aim of our studies is to investigate the natural effect on air quality, and the future work will focus on the air quality simulation to quantitatively investigate its effect on air pollution. This manuscript is the first step of this topic and we concentrated on the MEGAN model and its sensitivity to different inputs. Considering the comments from two reviewers, we will adjust the content of our manuscript, and the data and results from a recent published paper concentrated on same topic by Ren et al. (2017) would be added in the revised manuscript to further discuss this topic.

2) The methodology section needs to be significantly expanded to include more descriptions on experiment setup (see my specific comments below). Any novel settings should be highlighted.

Response: The authors thanks for your comments. We have replied the reviewer's specific comments below and will add more details of the configuration of the experiments in the revised manuscript and supplement.

- 3) The paper can benefit from careful language editing, preferably with help from a native English speaker. I recognize that the authors made some efforts to address this issue brought up during the ACPD quick report phase. However, the current version still contains many grammar mistakes and awkward sentences. References are often inaccurate/inappropriate. Transitions from one sentence to another, from one paragraph to another are not smooth. Also, when one paragraph ends and a new one begins, the authors should either indent the first line of the new paragraph, or leave a line space between the two paragraphs. Here are some suggested edits to the first sentences of your abstract.
- P1, L11: air quality pollution -> air pollution

P1, L12: delete "still", and also requires other emission inventories. A sentence saying BVOC emissions are sensitive to land and met conditions should be placed here.

P1, L15-16: "based on" -> "using"; add "the" before "Model of ....."; delete "model" after v2.1

P1, L19: "are used to design five experiments, as E1-E5, to calculate and test the sensitivity of the model" -> are used in five model sensitivity experiments, as E1-E5

P1, L20: "Based on the meteorological conditions from Weather Forecasting and Research (WRF) model, this inventory is an hourly inventory with 3-km spatial resolution."->These sensitivity calculations were driven by hourly, 3 km meteorological fields from the Weather Forecasting and Research (WRF) model.

Response: The authors thanks for your constructive suggestion. The language issue was mentioned by two reviewers and we would take some measures like inviting native speakers to help embellish the language, and the professional language editing has been called before the revised manuscript submitted.

Registered Office. Elsevier Ltd The Boulevard, Langford Lane Kidlington, OX5 1GB, UK, Registration No. 331566771

**To whom it may concern**

The paper "Sensitivity of Biogenic Volatile Organic Compounds Emissions to Leaf Area Index and Land Cover in Beijing" by Hui Wang, Qizhong Wu was edited by Elsevier Language Editing Services.

Kind regards,

**Elsevier Webshop Support**

**Specific suggestions and comments:**

1)Novelty: The authors argue that using spatially and temporally varying meteorological fields output from WRF is advantageous, compared with the approaches in some previous studies. Using WRF fields to drive MEGAN calculations is not at all a novel approach and has been widely used in a large number of studies, including some cited by the authors. As the authors are already aware, the uncertainty in their WRF simulation contributed to the estimated BVOC emission biases. There is no need to emphasize the benefit of driving MEGAN using WRF. Rather, if any novel configurations were applied to your WRF simulation, which helped reduce errors in the modeled T2, radiation, moisture, etc, they should be highlighted. See also my next comment.

Response: The authors appreciate your constructive comments. The authors agree to the reviewer's point of "driving the MEGAN model by mesoscale meteorological model is not a novel approach", indeed, multiple studies have adopted same methodology(Carlton and Baker, 2011;Shuping et al., 2010;Wang et al., 2011;Li and Xie, 2014). In this study, we emphasizing this part is to explain the different consideration of meteorological conditions compared with some previous studies. Compared with some previous studies (Klinger et al., 2002;Zhihui et al., 2003), this method could be the more reasonable way to explain the meteorological effect on the BVOCs emission. And the accuracy of the meteorological conditions is helpful to diminish the uncertainties of BVOCs emission from meteorological conditions. And we adopted default MODIS land cover provided by WRF-official group in our previous simulation. Therefore, considering reviewer's suggestion, we re-simulated the WRF model with updated land cover by using MODIS 12Q1 data for the summer in 2013. As showed in Table 1, the simulation results were validated by hourly in-situ temperature observation. The average R has a slight increase from 0.82 to 0.83, but the Root-Mean-Square-Error (RMSE) and Mean Error is *increased* from 2.67 °C and 3.34 °C to 3.07 °C and 3.70 °C, which means updating land cover of MODIS is not beneficial to improve the model performance under this situation. And the specific sites like Tong Zhou, Da Xing and Fang Shan still have the underestimation of temperature simulation, which could not be solved by the updating the land cover. Since the work mainly focus on the sensitivity of LAI and LC, the more effort would be paid on discussing the effect of LAI and LC inputs.

| Default MODIS land cover in WRF                                                                                                                                                            |                                                                                                                                      |                                                                                                                                                               |                                                                                                                                 |                                                                                                                  |  |  |
|--------------------------------------------------------------------------------------------------------------------------------------------------------------------------------------------|--------------------------------------------------------------------------------------------------------------------------------------|---------------------------------------------------------------------------------------------------------------------------------------------------------------|---------------------------------------------------------------------------------------------------------------------------------|------------------------------------------------------------------------------------------------------------------|--|--|
| ID                                                                                                                                                                                         | ME(°C)                                                                                                                               | MB(°C)                                                                                                                                                        | R                                                                                                                               | RMSE(°C)                                                                                                         |  |  |
| Beijing                                                                                                                                                                                    | 2.07                                                                                                                                 | 0.68                                                                                                                                                          | 0.83                                                                                                                            | 2.76                                                                                                             |  |  |
| Hai Dian                                                                                                                                                                                   | 2.16                                                                                                                                 | 0.93                                                                                                                                                          | 0.83                                                                                                                            | 2.93                                                                                                             |  |  |
| Chao Yang                                                                                                                                                                                  | 2.09                                                                                                                                 | -0.48                                                                                                                                                         | 0.82                                                                                                                            | 2.67                                                                                                             |  |  |
| Shun Yi                                                                                                                                                                                    | 2.02                                                                                                                                 | -0.35                                                                                                                                                         | 0.84                                                                                                                            | 2.61                                                                                                             |  |  |
| Huai Rou                                                                                                                                                                                   | 2.13                                                                                                                                 | 1.4                                                                                                                                                           | 0.84                                                                                                                            | 2.95                                                                                                             |  |  |
| Tong Zhou                                                                                                                                                                                  | 4.26                                                                                                                                 | -4.03                                                                                                                                                         | 0.78                                                                                                                            | 4.9                                                                                                              |  |  |
| Chang Ping                                                                                                                                                                                 | 2.3                                                                                                                                  | 0.75                                                                                                                                                          | 0.79                                                                                                                            | 3.19                                                                                                             |  |  |
| Yan Qin                                                                                                                                                                                    | 3.7                                                                                                                                  | 3.36                                                                                                                                                          | 0.79                                                                                                                            | 4.64                                                                                                             |  |  |
| Feng Tai                                                                                                                                                                                   | 2.31                                                                                                                                 | -1.19                                                                                                                                                         | 0.82                                                                                                                            | 2.88                                                                                                             |  |  |
| Shijing Shan                                                                                                                                                                               | 2.09                                                                                                                                 | -0.25                                                                                                                                                         | 0.81                                                                                                                            | 2.68                                                                                                             |  |  |
| Da Xing                                                                                                                                                                                    | 4.53                                                                                                                                 | -4.42                                                                                                                                                         | 0.81                                                                                                                            | 5.12                                                                                                             |  |  |
| Fang Shan                                                                                                                                                                                  | 3.9                                                                                                                                  | -3.63                                                                                                                                                         | 0.79                                                                                                                            | 4.48                                                                                                             |  |  |
| Mi Yun                                                                                                                                                                                     | 1.94                                                                                                                                 | 0.46                                                                                                                                                          | 0.83                                                                                                                            | 2.52                                                                                                             |  |  |
| Mentou Gou                                                                                                                                                                                 | 1.92                                                                                                                                 | 0.44                                                                                                                                                          | 0.83                                                                                                                            | 2.6                                                                                                              |  |  |
| Ping Gu                                                                                                                                                                                    | 2.57                                                                                                                                 | -1.77                                                                                                                                                         | 0.85                                                                                                                            | 3.1                                                                                                              |  |  |
| Avg                                                                                                                                                                                        | 2.67                                                                                                                                 | -0.54                                                                                                                                                         | 0.82                                                                                                                            | 3.34                                                                                                             |  |  |
| 8                                                                                                                                                                                          |                                                                                                                                      |                                                                                                                                                               |                                                                                                                                 |                                                                                                                  |  |  |
|                                                                                                                                                                                            | MOD                                                                                                                                  | IS 12Q1 land cover in                                                                                                                                         | 2013                                                                                                                            |                                                                                                                  |  |  |
| ID                                                                                                                                                                                         | MOD
MB(°C)                                                                                                                        | IS 12Q1 land cover in
ME(°C)                                                                                                                               | 2013
R                                                                                                                       | RMSE(°C)                                                                                                         |  |  |
| ID
Beijing                                                                                                                                                                              | MOD
MB(°C)
2.93                                                                                                                | IS 12Q1 land cover in
ME(°C)
2.64                                                                                                                       | 2013
R
0.85                                                                                                        | RMSE(°C)
3.59                                                                                                 |  |  |
| ID
Beijing
Hai Dian                                                                                                                                                                  | MOD
MB(°C)
2.93
3.23                                                                                                        | IS 12Q1 land cover in
ME(°C)
2.64
2.97                                                                                                               | 2013
R
0.85
0.83                                                                                                | RMSE(°C)
3.59
3.99                                                                                         |  |  |
| ID
Beijing
Hai Dian
Chao Yang                                                                                                                                                     | MOD
MB(°C)
2.93
3.23
1.96                                                                                                | IS 12Q1 land cover in
ME(°C)
2.64
2.97
0.02                                                                                                       | 2013                                                                                                                            | RMSE(°C)
3.59
3.99
2.53                                                                                 |  |  |
| ID
Beijing
Hai Dian
Chao Yang
Shun Yi                                                                                                                                          | MOD
MB(°C)
2.93
3.23
1.96
1.87                                                                                        | IS 12Q1 land cover in
ME(°C)
2.64
2.97
0.02
-0.22                                                                                              | 2013
R
0.85
0.83
0.83
0.83
0.85                                                                               | RMSE(°C)
3.59
3.99
2.53
2.41                                                                         |  |  |
| ID
Beijing
Hai Dian
Chao Yang
Shun Yi
Huai Rou                                                                                                                              | MOD
MB(°C)
2.93
3.23
1.96
1.87
3.67                                                                                | IS 12Q1 land cover in
ME(°C)
2.64
2.97
0.02
-0.22
3.56                                                                                      | 2013                                                                                                                            | RMSE(°C)   3.59   3.99   2.53   2.41   4.39                                                                      |  |  |
| ID
Beijing
Hai Dian
Chao Yang
Shun Yi
Huai Rou
Tong Zhou                                                                                                                 | MOD
MB(°C)
2.93
3.23
1.96
1.87
3.67
4.14                                                                        | IS 12Q1 land cover in
ME(°C)
2.64
2.97
0.02
-0.22
3.56
-3.92                                                                             | 2013
R
0.85
0.83
0.83
0.83
0.83
0.83
0.83
0.83                                                       | RMSE(°C)   3.59   3.99   2.53   2.41   4.39   4.73                                                               |  |  |
| ID
Beijing
Hai Dian
Chao Yang
Shun Yi
Huai Rou
Tong Zhou
Chang Ping                                                                                                   | MOD
MB(°C)
2.93
3.23
1.96
1.87
3.67
4.14
2.89                                                                | IS 12Q1 land cover in
ME(°C)
2.64
2.97
0.02
-0.22
3.56
-3.92
2.54                                                                     | 2013                                                                                                                            | RMSE(°C)   3.59   3.99   2.53   2.41   4.39   4.73   3.8                                                         |  |  |
| ID
Beijing
Hai Dian
Chao Yang
Shun Yi
Huai Rou
Tong Zhou
Chang Ping
Yan Qin                                                                                        | MOD
MB(°C)
2.93
3.23
1.96
1.87
3.67
4.14
2.89
5.34                                                        | IS 12Q1 land cover in
ME(°C)
2.64
2.97
0.02
-0.22
3.56
-3.92
2.54
5.33                                                             | R   0.85   0.83   0.83   0.83   0.83   0.84   0.83   0.83   0.83   0.83   0.83   0.83   0.83   0.83   0.83   0.83   0.81   0.8  | RMSE(°C)   3.59   3.99   2.53   2.41   4.39   4.73   3.8   6.04                                                  |  |  |
| ID
Beijing
Hai Dian
Chao Yang
Shun Yi
Huai Rou
Tong Zhou
Chang Ping
Yan Qin
Feng Tai                                                                            | MOD
MB(°C)
2.93
3.23
1.96
1.87
3.67
4.14
2.89
5.34
2.15                                                | IS 12Q1 land cover in
ME(°C)
2.64
2.97
0.02
-0.22
3.56
-3.92
2.54
5.33
-0.94                                                    | R   0.85   0.83   0.83   0.83   0.83   0.83   0.83   0.83   0.83   0.83   0.83   0.83   0.83   0.83   0.81   0.83               | RMSE(°C)   3.59   3.99   2.53   2.41   4.39   4.73   3.8   6.04   2.68                                           |  |  |
| ID
Beijing
Hai Dian
Chao Yang
Shun Yi
Huai Rou
Tong Zhou
Chang Ping
Yan Qin
Feng Tai
Shijing Shan                                                            | MOD
MB(°C)
2.93
3.23
1.96
1.87
3.67
4.14
2.89
5.34
2.15
2.35                                        | IS 12Q1 land cover in
ME(°C)
2.64
2.97
0.02
-0.22
3.56
-3.92
2.54
5.33
-0.94
1.39                                            | R   0.85   0.83   0.83   0.83   0.83   0.84   0.83   0.83   0.83   0.83   0.83   0.83   0.81   0.83   0.83   0.83               | RMSE(°C)   3.59   3.99   2.53   2.41   4.39   4.73   3.8   6.04   2.68   2.97                                    |  |  |
| ID
Beijing
Hai Dian
Chao Yang
Shun Yi
Huai Rou
Tong Zhou
Chang Ping
Yan Qin
Feng Tai
Shijing Shan
Da Xing                                                 | MOD
MB(°C)
2.93
3.23
1.96
1.87
3.67
4.14
2.89
5.34
2.15
2.35
4.47                                | IS 12Q1 land cover in
ME(°C)
2.64
2.97
0.02
-0.22
3.56
-3.92
2.54
5.33
-0.94
1.39
-4.38                                   | 2013                                                                                                                            | RMSE(°C)   3.59   3.99   2.53   2.41   4.39   4.73   3.8   6.04   2.68   2.97   5.01                             |  |  |
| ID
Beijing
Hai Dian
Chao Yang
Shun Yi
Huai Rou
Tong Zhou
Chang Ping
Yan Qin
Feng Tai
Shijing Shan
Da Xing
Fang Shan                                    | MOD
MB(°C)
2.93
3.23
1.96
1.87
3.67
4.14
2.89
5.34
2.15
2.35
4.47
3.89                        | IS 12Q1 land cover in
ME(°C)
2.64
2.97
0.02
-0.22
3.56
-3.92
2.54
5.33
-0.94
1.39
-4.38
-3.67                          | 2013                                                                                                                            | RMSE(°C)   3.59   3.99   2.53   2.41   4.39   4.73   3.8   6.04   2.68   2.97   5.01   4.46                      |  |  |
| ID
Beijing
Hai Dian
Chao Yang
Shun Yi
Huai Rou
Tong Zhou
Chang Ping
Yan Qin
Feng Tai
Shijing Shan
Da Xing
Fang Shan
Mi Yun                          | MOD
MB(°C)
2.93
3.23
1.96
1.87
3.67
4.14
2.89
5.34
2.15
2.35
4.47
3.89
1.93                | IS 12Q1 land cover in
ME(°C)
2.64
2.97
0.02
-0.22
3.56
-3.92
2.54
5.33
-0.94
1.39
-4.38
-3.67
0.78                  | R   0.85   0.83   0.83   0.83   0.83   0.83   0.83   0.83   0.83   0.83   0.83   0.81   0.82   0.82   0.83   0.82   0.83        | RMSE(°C)   3.59   3.99   2.53   2.41   4.39   4.73   3.8   6.04   2.68   2.97   5.01   4.46   2.48               |  |  |
| ID
Beijing
Hai Dian
Chao Yang
Shun Yi
Huai Rou
Tong Zhou
Chang Ping
Yan Qin
Feng Tai
Shijing Shan
Da Xing
Fang Shan
Mi Yun
Mentou Gou            | MOD
MB(°C)
2.93
3.23
1.96
1.87
3.67
4.14
2.89
5.34
2.15
2.35
4.47
3.89
1.93
2.67        | IS 12Q1 land cover in
ME(°C)
2.64
2.97
0.02
-0.22
3.56
-3.92
2.54
5.33
-0.94
1.39
-4.38
-3.67
0.78
2.14          | R   0.85   0.83   0.83   0.83   0.83   0.83   0.83   0.83   0.83   0.83   0.83   0.83   0.81   0.82   0.82   0.82   0.85   0.82 | RMSE(°C)   3.59   3.99   2.53   2.41   4.39   4.73   3.8   6.04   2.68   2.97   5.01   4.46   2.48   3.36        |  |  |
| ID
Beijing
Hai Dian
Chao Yang
Shun Yi
Huai Rou
Tong Zhou
Chang Ping
Yan Qin
Feng Tai
Shijing Shan
Da Xing
Fang Shan
Mi Yun
Mentou Gou
Ping Gu | MOD
MB(°C)
2.93
3.23
1.96
1.87
3.67
4.14
2.89
5.34
2.15
2.35
4.47
3.89
1.93
2.67
2.6 | IS 12Q1 land cover in
ME(°C)
2.64
2.97
0.02
-0.22
3.56
-3.92
2.54
5.33
-0.94
1.39
-4.38
-3.67
0.78
2.14
-1.92 | R   0.85   0.83   0.83   0.83   0.83   0.83   0.83   0.83   0.83   0.83   0.81   0.82   0.82   0.85   0.85                      | RMSE(°C)   3.59   3.99   2.53   2.41   4.39   4.73   3.8   6.04   2.68   2.97   5.01   4.46   2.48   3.36   3.13 |  |  |

Table 1. The meteorological validation with in-situ observation. MB, ME and RMSE is the abbreviation of Mean Error, Mean Bias and Root-Mean Square Error, and the *R* is the correlation coefficient.

2)More information regarding your WRF simulation and evaluation approach should be provided in Section 2.2.1. These should include:

Response: The authors are grateful for your valuable suggestions. As mentioned above, we have supplemented the description about configuration and evaluation of the model in the revised manuscript. And corresponding questions or comments would be replied point to point as following:

**-introduce the initialization time for each domain.**

Response: The model is initialized at 12:00 UTC every day, and initial and boundary conditions are provided by the FNL(Final) Operational Global Analysis data(National Centers for Environmental Prediction, 2000); the boundary conditions are updated every 6 hours. The first 12 hours are treated as the spin-up time. And we cut and merged the medium 24 hours, from 00:00 A.M. UTC to 23:00 P.M. UTC, to drive the MEGAN model to estimate the BVOCs emission.

**-introduce the vertical spacing for each domain.**

Response: The three domains in model all contain same 27 vertical layers above the ground and 4 vertical layers under the ground, and the sigma values of model vertical layers are:

1, 0.993, 0.983, 0.97, 0.954, 0.934, 0.909, 0.88, 0.8295757, 0.7791514,0.7287272, 0.6783029, 0.5917439, 0.5136936, 0.4434539, 0.3803751, 0.3238531, 0.2733261, 0.228273, 0.18821, 0.1526888, 0.1212943,0.09364247, 0.0693781, 0.04817315, 0.02972473, 0.01375316, 0.

-introduce your physics options for each domain and their suitability for the Beijing area (based on literature and/or any sensitivity simulations the authors may have conducted)

Response: The authors appreciate your previous comments. The physical options would be presented in the supplement.

-introduce the land cover and vegetation dynamics (e.g., green vegetation fraction) input data, including the year of these input data represented, and discuss how they may have contributed to biases from your WRF simulation.

Response: The land cover of WRF simulation was using default MODIS land cover datasets. The discussion of impact on meteorological simulation by updating the land cover or vegetation fraction is focused on the WRF simulation, which is out of the scope of this study mainly concentrated on. In addition, we also did the sensitivity simulation to test the impact of updating the land cover by the MODIS 12Q1 land cover data in 2013, and hourly temperature validation indicated that such measure didn't significantly help to improve the meteorological simulation. The further discussion about the optimize physical schemes or parameters may out of the object of this study, and the meteorological validation has demonstrated the reasonability of our meteorological simulation on the key condition like temperature and radiation.

- P4, L23: Skamarock et al., 2005 is for WRF version 2. Please cite WRF version 3 documentation.

Response: We have modified this reference from Skamarock et al. (2005) to Skamarock et al. (2008).

- P4, L27: justify "we considered the second day as the reasonable results", for example, compare the 1-day and 2day model performance.

Response: As mentioned above, the 2-day simulations of WRF were done day by day, and the model is initialized at 12:00 UTC. The simulation lasts 48 hours and first 12 hours are as the spin-up time. The data of the period from 00:00 A.M. UTC to 23:00 P.M. UTC of the second day was cut and collapsed to estimate the BVOCs emission. The "reasonable" part means the simulation without the spin-up time, but such configuration was not expressed clearly by this sentence. Therefore, we modified this paragraph as following to verify the processing of WRF data:

"The WRF model was initialized at 12:00 UTC, and the first 12 hours were spin-up time. The data of the period from 00:00 A.M. UTC to 23:00 P.M. UTC in the second day was cut and merged to estimate the BVOC emissions. The merged file was processed by the Meteorology-Chemistry Interface Processor (MCIP) (Otte and Pleim, 2010) tool to provide meteorological conditions for MEGAN model."

- P4, L29: explain why daily T2 was evaluated, instead of hourly T2? Change "among" to "within";

Response: Considering the observation of radiation is daily, in order to evaluate the two variables at the same level, daily T2 was used to evaluate the simulation ability of model to daily variance of whole year meteorological conditions. We also did the validation with the hourly T2 as presented in Table 2. The validation results are similar

with the previous validation with daily temperature. Therefore, there is no obvious difference to use daily or hourly temperature data to validate the simulation and they all illustrate the reasonability of our meteorological simulation. Table 2. The meteorological validation with hourly T2 in-situ observation.

| ID           | ME(°C) | MB(°C) | R    | RMSE(°C) |
|--------------|--------|--------|------|----------|
| Beijing      | 1.9    | -0.13  | 0.97 | 2.48     |
| Hai Dian     | 2      | 0.1    | 0.97 | 2.67     |
| Chao Yang    | 2.35   | -1.07  | 0.96 | 2.91     |
| Shun Yi      | 2.32   | -1.3   | 0.97 | 2.93     |
| Huai Rou     | 2.12   | 0.28   | 0.96 | 3.13     |
| Tong Zhou    | 4.91   | -4.77  | 0.96 | 5.5      |
| Chang Ping   | 2.03   | -0.4   | 0.97 | 2.73     |
| Yan Qin      | 3.11   | 1.88   | 0.94 | 4.3      |
| Feng Tai     | 2.9    | -2     | 0.96 | 3.55     |
| Shijing Shan | 2.34   | -0.98  | 0.96 | 2.86     |
| Da Xing      | 5.22   | -5.04  | 0.95 | 5.92     |
| Fang Shan    | 4.73   | -4.44  | 0.94 | 5.6      |
| Mi Yun       | 2.61   | -0.36  | 0.94 | 3.66     |
| Mentou Gou   | 2.01   | -0.5   | 0.97 | 2.57     |
| Ping Gu      | 3.3    | -2.53  | 0.95 | 4.25     |
| Avg          | 2.92   | -1.42  | 0.96 | 3.67     |

- P5, L1/L4: unit is missing for these biases. Why was the MB of -1.5 degree mentioned twice?

Response: Thanks for your precious comments. We have followed the reviewer's suggestion and added the unit of biases in the article. The MB mentioned twice is for emphasizing the general cooling bias of the temperature simulation.

- P5, L9: which single station?

Response: It's Beijing Station, No. 54511. The information will be added in the revised manuscript.

3) Issues regarding satellite products:

- P6, L22: "Because of the highest spatial resolution of the FROM LC product, the experiment using FROM PFT and GLASS LAI as inputs is the baseline experiment (E1)". I don't understand the logical connections between resolution and choice of the baseline experiment.

Response: The authors appreciate your precious comments. According to the description of MEGAN(Guenther et al., 2012), the emission factor is decided by the distribution of Plant Function Types (PFTs). The sub-grid categories in the specific gird are presented by calculating the area fraction of different PFTs, which means the high-resolution can provide more details of PFTs distribution and calculate more accurate fractions of PFTs. Therefore, we treated the Fine Resolution Observation and Monitoring of Global Land Cover (FROM-GLC) land cover datasets with 30m spatial resolution as the baseline experiment.

- Although the land cover datasets used in this study differed by at least a factor of ten in resolution (30m vs 500/300m), they are all at much finer resolution than the 3 km WRF-MEGAN grid. It would be helpful to explain how these data were regridded to your WRF-MEGAN model grid. This would help us understand how the original data resolution may have affected your results. Approach used to reproject the original 1 km LAI data should also be provided. Missing a "respectively" in P5, L25.

Response: The authors agree with the reviewer. The WRF-MEGAN grid is coarser than the land cover grids, and we used the Preprocessing Tools of MEGAN to regrid the grids by calculating the area fractions of different landscapes

or PFTs) in WRF grid. Furthermore, the original LAI data was also regridded to WRF-MEGAN grid through the calculating the area mean LAI.

- P5, Section 2.2.2: The land cover and LAI data citations are not helpful. For each dataset, please cite the corresponding algorithm/validation paper, and provide the dataset doi or/and accurate links to retrieve the data. MOD15 is not an accurate description for the used MODIS LAI data. I assume the correct format should be MCD15A2H, Collection/Version XX. Same issue exsits in Table 4.

Response: The authors thank your constructive comments. We have cited the relevant papers and added the doi of the datasets to help readers to retrieve the data, and the name of MODIS LAI data has been modified. The availability of datasets and code is added as an independent section in revised manuscripts as following:

"The source code of WRF model V3.3.1 and MEGAN v2.1 is available at http://www2.mmm.ucar.edu/wrf/users/ and https://bai.ess.uci.edu, respectively. The FROM-GLC can be downloaded from the website of Department of Earth System Science, Tsinghua University, at http://data.ess.tsinghua.edu.cn/index.html. The CCI-LC can be downloaded from the website of Climate Change Initiative Program at https://www.esa-landcover-cci.org. The GLASS LAI can be obtained through the website of National Earth System Science Data Sharing Infrastructure at http://www.geodata.cn/thematicView/GLASS.html or the website of Global Land Cover Facility, University of Maryland, at http://glcf.umd.edu/data/lai/. The GEO v2 LAI is available on the website of the Copernicus Global Land Service at https://land.copernicus.eu/global/products/. The MODIS MCDQ12 LC and MODIS MCD15A2 LAI, Version 5, are available on the website of Land Process Distributed Active Center at https://lpdaac.usgs.gov/dataset\_discovery/modis/modis\_products\_table."

- Did the author screen the LAI data and if so, based on what criteria? What values were used for grids with missing data/unrealistic (e.g., extremely high) LAI? Previously studies have reported MEGAN sensitivity to PFT and LAI, so it'd be helpful to compare your findings with theirs.

Response: The authors appreciate your comments. In this study, the LAI datasets are the level 4 satellite products, and the GEOv2 as well as GLASS LAI products have adopted some measures to remove and fill the unrealistic or unreasonable value (Verger et al., 2014;Xiao et al., 2014). The MODIS LAI products adopting vegetation canopy radiation models of diverse plants type to produce LAI products, and if the canopy model is not available for the pixel, the backup algorithm of estimating LAI by using Normalized Difference Vegetation Index (NDVI) would be used (Knyazikhin et al., 1999). And the MODIS products also provided the quality flags to distinguish the quality of products, and the remain missing values are the pixel mixed with no-vegetation types like water. Considering the characters of the MODIS LAI, we checked all the available values to make sure that they are in reasonable range of LAI (0-7), and used all available values. We didn't use interpolating method to fill the missing value to avoid extra uncertainty and will further compare the effect of LAI products by considering two aspects, masking area and LAI value discrepancy. And we will follow the reviewer's suggestion to compare our results with other publications.

5) The uncertainty section (3.5) is not well written, and the current discussions are very qualitative and not informative.

Response: The authors thank for your comments. We have removed this section and added more informative as well as quantitative results in the Discussion section.

6) P2, L25-27: Please provide the source for "the statistical data from the Nation Forest Resources Survey (NFRS) reported that the forest coverage rate in Beijing rose from 20.6% to 35.8% during 1998-2013". This is also the right place to mention the impact of different met conditions during the earlier periods and 2013.

Response: The authors thank for the reviewer's comments. The data of forest coverage rate of China and specific provinces came from the website of the China Forestry Database (http://data.forestry.gov.cn/lysjk). We also followed the met conditions effect on this period at the same position.

7) P1, L14; P2, L28; P3, L1: "new" is not accurate. Previous estimates of BVOCs emissions introduced by the authors are not for 2013.

Response: Thanks for your comments. We have modified it in the revised paper and removed the word "new". 8) P5, L25: Shouldn't the last sentence belong to Section 2.2.1? Define MCIP, and use the correct link for MCIP. Response: The authors appreciate your comments. We have move the last sentence of this paragraph to Section 2.2.1, and the abbreviation of MICP was extended to the full name, Meteorology-Chemistry Interface Processer, with citing the corresponding paper from Otte and Pleim (2010).

8) P5, L28: four species -> four groups

Response: Thanks for your precious comments. We have followed the review's comments and modified this error. 9) P9, Section 3.3: Method of this sensitivity test should be first introduced in Section 2.

Response: Thank you for the comments. The introduction of the contribution calculation has been moved to supplement of the manuscript.

10) To comply with the ACP policy, data availability should be included in the "Acknowledgements" section. Response: Thank you for the comments. We have followed the reviewer's comments and added data availability in the revised manuscript as following:

"The source code of WRF model V3.3.1 and MEGAN v2.1 is available at http://www2.mmm.ucar.edu/wrf/users/ and https://bai.ess.uci.edu, respectively. The FROM-GLC can be downloaded from the website of Department of Earth System Science, Tsinghua University, at http://data.ess.tsinghua.edu.cn/index.html. The CCI-LC can be downloaded from the website of Climate Change Initiative Program at https://www.esa-landcover-cci.org. The GLASS LAI can be obtained through the website of National Earth System Science Data Sharing Infrastructure at http://www.geodata.cn/thematicView/GLASS.html or the website of Global Land Cover Facility, University of Maryland, at http://glcf.umd.edu/data/lai/. The GEO v2 LAI is available on the website of the Copernicus Global Land Service at https://land.copernicus.eu/global/products/. The MODIS MCDQ12 LC and MODIS MCD15A2 LAI, Version 5, are available on the website of Land Process Distributed Active Center at https://lpdaac.usgs.gov/dataset discovery/modis/modis products table."

11) Captions of Figures 4, 6, 7: specify which experiment these were based on.

Response: Thank you for the comments. We would modify these figures and make them more clear and informative.

**Reference**

Carlton, A. G., and Baker, K. R.: Photochemical modeling of the Ozark isoprene volcano: MEGAN, BEIS, and their impacts on air quality predictions, Environ Sci Technol, 45, 4438-4445, 10.1021/es200050x, 2011.

Guenther, A. B., Jiang, X., Heald, C. L., Sakulyanontvittaya, T., Duhl, T., Emmons, L. K., and Wang, X.: The Model of Emissions of Gases and Aerosols from Nature version 2.1 (MEGAN2.1): an extended and updated framework for modeling biogenic emissions, Geoscientific Model Development, 5, 1471-1492, 10.5194/gmd-5-1471-2012, 2012. Klinger, L. F., Li, Q. J., Guenther, A. B., Greenberg, J. P., Baker, B., and Bai, J. H.: Assessment of volatile organic compound emissions from ecosystems of China, Journal of Geophysical Research: Atmospheres, 107, ACH 16-11-ACH 16-21, 10.1029/2001jd001076, 2002.

Knyazikhin, Y., Glassy, J., Privette, J. L., Tian, Y., Lotsch, A., Zhang, Y., Wang, Y., Morisette, J. T., P.Votava, Myneni, R. B., Nemani, R. R., and Running, S. W.: MODIS Leaf Area Index (LAI) and Fraction of Photosynthetically Active Radiation Absorbed by Vegetation (FPAR) Product (MOD15) Algorithm Theoretical Basis Document, 1999.

Li, L. Y., and Xie, S. D.: Historical variations of biogenic volatile organic compound emission inventories in China, 1981–2003, Atmospheric Environment, 95, 185-196, https://doi.org/10.1016/j.atmosenv.2014.06.033, 2014.

Otte, T. L., and Pleim, J. E.: The Meteorology-Chemistry Interface Processor (MCIP) for the CMAQ modeling system: updates through MCIPv3.4.1, Geosci. Model Dev., 3, 243-256, 10.5194/gmd-3-243-2010, 2010.

Ren, Y., Qu, Z., Du, Y., Xu, R., Ma, D., Yang, G., Shi, Y., Fan, X., Tani, A., Guo, P., Ge, Y., and Chang, J.: Air quality and health effects of biogenic volatile organic compounds emissions from urban green spaces and the mitigation strategies, Environmental Pollution, 230, 849-861, https://doi.org/10.1016/j.envpol.2017.06.049, 2017.

Shuping, S., Xuemei, W., GUENTHER, A., CHEN, F., Ziwei, C., Zhiyong, W., and Zhiqin, L.: Impacts of errors in meteorological simulations on estimation of isoprene emission, Acta Scientiae Circumstantiae, 30, 2383-2391, 2010. Skamarock, W. C., Klemp, J. B., Dudhia, J., Gill, D. O., Barker, D. M., Wang, W., and Powers, J. G.: A description of the advanced research WRF version 2, NCAR Technical Note NCAR/TN-468+STR, 2005.

Skamarock, W. C., Klemp, J. B., Dudhia, J., Gill, D. O., Barker, D. M., Duda, M. G., Huang, X.-y., Wang, W., and Powers, J. G.: A description of the advanced research WRF version 3, NCAR Technical Note NCAR/TN-475+STR, 2008.

Verger, A., Baret, F., and Weiss, M.: Near Real-Time Vegetation Monitoring at Global Scale, IEEE Journal of Selected Topics in Applied Earth Observations and Remote Sensing, 7, 3473-3481, 10.1109/JSTARS.2014.2328632, 2014.

Wang, X., Situ, S., Guenther, A., Chen, F. E. I., Wu, Z., Xia, B., and Wang, T.: Spatiotemporal variability of biogenic terpenoid emissions in Pearl River Delta, China, with high-resolution land-cover and meteorological data, Tellus B, 63, 241-254, 10.1111/j.1600-0889.2010.00523.x, 2011.

Xiao, Z., Liang, S., Wang, J., Chen, P., Yin, X., Zhang, L., and Song, J.: Use of General Regression Neural Networks for Generating the GLASS Leaf Area Index Product From Time-Series MODIS Surface Reflectance, IEEE Transactions on Geoscience and Remote Sensing, 52, 209-223, 10.1109/tgrs.2013.2237780, 2014.

Zhihui, W., Yuhua, B., and Shuyu, Z.: A biogenic volatile organic compounds emission inventory for Beijing, Atmospheric Environment, 37, 3771-3782, https://doi.org/10.1016/S1352-2310(03)00462-X, 2003.

---

## Author Response (AR1)

**Dear editor,**

Thanks a million for your precious time. According to the two reviewers' comments, we modified the manuscripts at these main aspects:

1) We updated the emission factors according to Klinger et al. (2002), Wang et al. (2003) and Ren et al. (2017), and the results were updated and compared with the original work in Ren et al. (2017) to discuss the uncertainty.

2) We compressed the descriptions about the MEGAN model results and focused on the discussion about the sensitivities of diverse inputs. In the revised manuscript, the Section 3.3, "The Roles of Different PFTs of BVOC Emissions in Beijing" and the Section 3.5, "Uncertainties", were removed and more informative contents were added in the Discussion. The Discussion section was divided into two sub-sections: "Sensitivity of BVOC emissions to LAI and PFT" and "Comparison with previous studies". Two more sensitivity experiments were added to discuss the impact of the cross-walking table for converting land cover classes to PFTs, and corresponding results were added to the supplement.

3) Since two reviewers both mentioned the language issue, the professional language editing from Elsevier corporation has been called before the revised manuscript submitted. We hope this measure could further improve the presentation quality of revised manuscript.

[Figure]

**Language Editing Services**

*Registered Office:*
Elsevier Ltd
The Boulevard, Langford Lane,
Kidlington, OX5 1GB, UK.
Registration No. 331566771

**To whom it may concern**

The paper "Sensitivity of Biogenic Volatile Organic Compounds Emissions to Leaf Area Index and Land Cover in Beijing" by Hui Wang, Qizhong Wu was edited by Elsevier Language Editing Services.

Kind regards,

**Elsevier Webshop Support**

According to the reviewer specific comments, we finished the response and modification point to point.

**To Referee #1:**
**General comment**

1) This manuscript describes a modeling study of biogenic VOC (BVOC) emissions from Beijing China. Since BVOC emissions are important for determining atmospheric composition and chemistry and are not well understood, this original study has the potential to contribute to the scientific understanding on this significant topic. The manuscript is difficult to understand in many places but that could be addressed with a thorough language editing.

Response: The authors appreciate your precious time and comments. As mentioned in the manuscript, this study is to investigate and discuss the sensitivities of MEGAN model using multiple satellite-based datasets. Therefore, it is necessary to evaluate the effect from input data on the estimation of BVOC emissions. The professional language editing has been called before the revised manuscript submitted to solve the language issue.

[Figure]

**Language Editing Services**

*Registered Office:*
Elsevier Ltd
The Boulevard, Langford Lane,
Kidlington, OX5 1GB, UK.
Registration No. 331566771

**To whom it may concern**

The paper "Sensitivity of Biogenic Volatile Organic Compounds Emissions to Leaf Area Index and Land Cover in Beijing" by Hui Wang, Qizhong Wu was edited by Elsevier Language Editing Services.

Kind regards,

**Elsevier Webshop Support**

2) The authors apply the MEGAN model, driven with WRF meteorology as is typically the case for MEGAN simulations. The most valuable part of the study is the investigation of the two of the main drivers of BVOC emissions: meteorology and landcover. For meteorology, the authors compare temperature to observations and report a bias of cool temperature simulated by the model that is likely because the model does not adequately simulate the impact of the "urban heat island" on temperature. This is one of the more interesting results of the study and is a topic that the authors could potentially explore further with a more detailed examination and discussion of the canopy and leaf temperature simulations. For land cover, the authors compare different satellite based datasets. They do not compare with any in-situ observations so it is a sensitivity study with limited insights regarding accuracy and uncertainties.

Response: The authors appreciate the reviewer's comments. Regarding with meteorology conditions, in the manuscript, we did some validation to indicate the reasonability of simulation to estimate the BVOCs emission. The reviewer mentioned that exploring the impact of Urban Heat Island (UHI) may be an interesting topic, however, the most of available satellite-based land cover datasets, like MODIS 12Q1, can't present the green land among the city region for the limitation of spatial resolution. In this study, only the Finer Resolution Observation and Monitoring of Global Land Cover (FROM-GLC) with 30m resolution could primarily characterize the major green land in the city, but the scattered green space like road greening can't be recognized, which make it not wise enough to discover this topic. The field surveys can provide the more thorough data of urban green space and the multiple studies have adopted such method to discover the relevant topic (Ren et al., 2017;Ghirardo et al., 2016;Chang et al., 2012), and our study focuses on the typical landscape, which could be distinguished in the satellite land cover products.

The validations of the land cover datasets were done by the land-cover validation dataset from Zhao et al. (2014), and the baseline period of validation dataset is 2009-2011. The 61 available sample points from the above dataset in Beijing and its surrounding regions were used and the results are showed in Table 1. The validation showed that the

FROM-GLC has the highest accuracy of 59.67% among these land cover datasets. Since FROM LC has the same benchmark period with validation dataset and close spatial resolution, the FROM LC showed the better accuracy than the other two products. Considering effect of the spatial resolution and benchmark time, the validation results only indicate the reasonability of the land covers coarsely.

Table 1. Accuracies calculated based on the sample points from land-cover validation dataset.

|  | FROM-GLC | MODIS 12Q1 2013 | CCI LC 2013 |
| --- | --- | --- | --- |
| Accuracy | 59.67% | 54.10% | 50.81% |

And the corresponding modified contents were added to page 10, line 26 in the revised manuscript as following:

"The BVOC emission results from the MEGAN model are more sensitive to the PFTs than to the LAI. Which LC dataset is used in the model significantly affects the BVOC emission estimates (Zhao et al., 2016; Wang et al., 2011). There are two major sources of uncertainty in the PFTs: the accuracy of the LC map and the cross-walking table used to convert the LC classes to PFTs (Hartley et al., 2017). 61 sample points in Beijing and the surrounding area from the Land Cover Validation Dataset by Zhao et al. (2014) were used as primary validation for the LC data sets. The validation samples were collected from TM/ETM images for 2009–2011. The accuracies of the FROM, MODIS, and CCI-LC datasets are 59.67%, 54.1%, and 50.81%, respectively. Since FROM LC has the same benchmark period as the validation dataset and similar spatial resolution, the FROM LC displayed better accuracy than the other two products. The validation results only can only coarsely assess the accuracy of the LC data sets. The advantage of the high-resolution data is that it diminishes the uncertainties associated with mixed pixels in the medium-resolution LC map, which relies on the cross-walking table to convert the LC classes to PFTs."

**Specific suggestions and comments:**

1) Conclusions #1 to 3 and #6 relate to the total emissions and the contribution of individual seasons. This would be of more interest if the study included some comparisons to BVOC emission measurements, so we have some idea if the emission results are correct. Since the paper does not include any observations of BVOC emissions, the MEGAN predictions of Beijing emissions should not themselves be the major focus of the manuscript. The current manuscript text (in the conclusion and elsewhere) devoted to describing the MEGAN model results (totals, seasonal and spatial variations) is too long and should be provided only as a brief description in the manuscript, and could perhaps be included in more detail in a supplemental section.

Response: Thanks for your comments. We noticed that BVOCs observations that can be found in previous publication (Shao et al., 2009;Xie et al., 2008;Wu et al., 2016;Li et al., 2015) are mainly for the sites in Peking University (PKU) and Yufu, a city site and a rural site. These publications only provide the average value and standard deviation, but are not for year 2013. To evaluate the BVOCs emission and effect, the chemistry transport model (CTM) should be used (Geng et al., 2011;Zhao et al., 2016), but there is no time-serious observation of BVOCs in 2013, and the observation sites are not located in the forest region. According to the reviewer's comments, we partly decreased the descriptions of model results and added more details of the discussion of the sensitivity tests. Meanwhile, we adjusted the emission factor based on Ren et al. (2017) to discuss and compare the results of two similar studies.

The modifications about adjusting emission factor of PTFs were in page 4, line 1 of the revised manuscript as following:

"Based on the area data of diverse tree species from the eighth NFRS (Table S1) which came from Ren et al. (2017), the emission factors of isoprene for the PFTs mentioned above are calculated as

$$\varepsilon_j = \sum \varphi_i LMA_i \frac{s_i}{s_j}$$

where $\varepsilon_j$ is the canopy-scale emission factor of the PFT species j, $\varphi_i$ is the leaf-scale emission factor for the vegetation species i, and the $S_i/S_j$ represents the area proportion of vegetation species i in PFT j."

The modifications about the comparison of two studies were in page 12, line 16 of the revised manuscript as following:

"The estimates made by Ren et al. (2017) used a species-level vegetation inventory based on field surveys, while this study used PFT-scale estimates based on satellite datasets, with statistically-derived PFT emission factors. The former method may be more accurate than the latter since emission factors differ between tree species, but it is limited by the rough process of data collection, and satellite based inventories are more easily gridded, facilitating coupling with chemistry transport models and thus allowing further investigation of the effect of BVOCs on atmospheric chemistry. Moreover, in Ren et al. (2017) the coverage of broadleaf trees and needleleaf trees is 18.7% and 7.4%, respectively, and in this study, it is 20.0%−30.0% and 6.3%−7.3%, respectively; i.e., the coverage of needleleaf trees, the main contributors to the emission of monoterpenes, is similar between the two studies, and thus the monoterpene emissions are relatively consistent. However, the isoprene estimates in this study are generally higher than those of Ren et al. (2017): the ratios of isoprene emissions between the two studies is 1.15−1.7, which are similar to the ratios (1.06−1.6) of broadleaf tree area between the two studies."

2) Conclusions #4 (LAI) and 5 (PFT) are the potentially more interesting contributions. However, there are several issues regarding the results and associated conclusions. Page 11, line 24/25 states that MODIS LAI led to a 17.4% decline of total BVOCs compared with baseline in this study, because of the relatively big mask area in the MODIS LAI product. This is not a reasonable comparison. The mask indicates that no data is being provided for the masked region so it doesn't make sense to compare them. The default MEGAN LAI data on the MEGAN website replaces the MODIS LAI in the masked region with values based on an interpolation from the surrounding region. You could use this or some other approach but it is misleading to indicate that the MODIS LAI is lower as indicated in the conclusion section and elsewhere (e.g., Figure3).

Response: Thanks for your comments. We have further compared the effect of LAI products by considering two aspects, masking area and LAI value discrepancy. Firstly, we compared the regions that is available in MODIS LAI products, and it could explain the effect from the discrepancy of the LAI value on the BVOCs emission estimation. In addition, according to the Xiao et al. (2016), the direct validation with in-situ observation shows that the GLASS LAI has most similar results with the site observation, and the MODIS LAI is the worst. Secondly, the BVOCs discrepancy from the masking area of MODIS LAI is isolated. Since the MODIS adopting the vegetation canopy radiation model to produce LAI products (Knyazikhin et al., 1999), the region assigned as non-pure vegetation type would be treated as a missing value. Furthermore, in this study, we used L4 level satellite products, and the producer of datasets has finished the work of interpolating the missing value in a reasonable range. Therefore, adopting the method like interpolating for the missing value in MODIS LAI is not helpful to improve the quality of the datasets but lead into new source of error, and we separately discussed the discrepancy within MODIS and other LAI products. The modification about the impact on the quantity of BVOC emissions of LAIs were added to page 8, line 3 of revised manuscript as following:

"E2 and E4 have similar total emissions at 75.7 Gg and 76.5 Gg, respectively, while E3 and E5 have lower emissions at 61.8 Gg and 56.0 Gg, respectively. The GEO LAI total emissions are more similar to the E1 results than those of the MODIS LAI. However, if only grid cells over which the MODIS LAI has no missing values are taken into account, the total BVOC emissions of E1-E3 are 63.5 Gg, 62.6 Gg, and 61.8 Gg; i.e., the MODIS MCD15A2 LAI and the GEO LAI lead to 1.4% and 2.6% difference with E1, respectively. The problem is that the E1 BVOC emissions in the region where the MODIS LAI has missing values account for 16.3% of the total E1 emissions. Considering the importance of BVOC emissions in suburban areas on air quality, the GEO and GLASS LAI may be

better choices for use in BVOC estimation for regional air quality simulation and forecasting. In particular, the estimates obtained using the GEO LAI for specific BVOC species all differ from E1 by less than 4%."

The discussion about the impact on BVOC emissions from LAIs were also added to page 10, line 17 of revised manuscript as following:

"To study the effect of the LAI input on BVOC emissions, we adopted three independent satellite-derived LAI datasets. According to the direct validation by Xiao et al. (2016), the GLASS and the GEO LAI are generally of better quality than the MODIS MCD15A2 LAI. Although the average MODIS MCD15A LAI is lower than the GLASS and GEOv2 LAI, the comparison of BVOC emissions with E1 in the region over which MODIS is valid (i.e., no missing values) showed that use of the GEOv2 and MODIS LAI input led to decreases of only 1.4% and 2.6%, respectively. The estimate of BVOC emissions in Beijing does not appear to be sensitive to the LAI input. However, considering the missing values in the MODIS MCD15A2 LAI, using the GLASS LAI and the GEO LAI is a better solution than using interpolation to fill in the missing values in the MODIS LAI."

3) Page 11, line 27/28 The statement, "Generally, the uncertainty of LAI is limited under the MEGAN model frame", is unclear but seems to suggest that because the GEO and GLASS LAI data products are similar that means that LAI uncertainties do not contribute substantially to MEGAN BVOC emission uncertainties. This is not necessarily the case as it probably just shows that the two datasets are based on a similar approach (with similar errors).

Response: Thanks for your comments. The authors have followed the reviewer's comments. And this statement has been deleted from the manuscript.

4) Regarding conclusion #5, and the PFT comparison in general, the authors apparently consider only the relative contribution of PFTs to the vegetation covered regions and do not consider the differences in total vegetation cover. I assume this is the case since the PFTs in table 3 add up to 100% but I expect the vegetation cover in Beijing must be less than 100%. How does total vegetation cover differ between the three landcover databases? In addition, the conclusion #5 reports the PFT cover differences but does not provide any insights on which is the most accurate, how uncertain they are, and what the implications are for modeling. For example, how important is it to get the relative PFT correct in comparison to getting total vegetation cover correct or accounting for the variability of emission factors within each PFT (i.e., not all broadleaf trees have the same isoprene emission factor).

Response: Thanks for your comments. The data in table 3 has been corrected to the fractions of area of the different land covers to the total area of the Beijing region. The vegetation distribution determined the standard emission factor. Satellite-based land cover products could provide the gridded spatial distribution of major landscapes, but it is limited to provide the further detail of the species information of different vegetation. Therefore, this approach is not suitable to solve the problem, which is mentioned by the reviewer that different species with same PFT have diverse emission factors. But the results based on satellite-based land cover map are gridded and available for the chemistry model directly, and such method is also the most common way for the researches of adopting CTM to investigate the topics about the air quality (Gao et al., 2016;Situ et al., 2013;Wei et al., 2018).

On the other hand, the field surveys can provide more information of species compositions, but the accurate spatial distribution of species is not available. The work done by Ren et al. (2017), mentioned by the reviewer, adopted the statistic species data from field surveys at administrative-region scale. And this approach may be more thorough to estimate the total BVOCs emission, but such method still faces the problem how to realize gridding of these results and make them suitable for the CTM.

In general, adopting the satellite-based PFT map would lead to the errors from the species diversity of emission factor, but it is easier to be gridded for the following research. A compromising way is to estimate the emission

factor of PFTs based on the statistical data of vegetation species, which presents the average emission factor of the PFTs. And this method was also adopted by Wang et al. (2011) to provide reasonable parameters for the estimation of regional BVOCs emission.

In addition, the classification of the satellite land cover is also play a key role in determining the emission factor. The PFTs scheme in MEGAN v2.1 are from Community Land Model v4 (CLM4)(Lawrence et al., 2011), and it is significant to convert the diverse land cover classes to the PFTs, which is called cross-walking. The MODIS MCD12Q1 LC and the Climate Change Initiative Land Cover (CCI-LC) adopt the corresponding cross-walking tables to convert its classification system to PFTs. The FROM-GLC product doesn't provide the corresponding cross-walking table, therefore, the converting of FROM-GLC is based on the legend of its original classification system. Since pixels of the medium-resolution land cover datasets would contain the information of multiple land cover types and that the cross-walking table is one of the sources of uncertainty in land surface model (Hartley et al., 2017), we treated the 30m resolution FROM-GLC as the baseline land cover in the experiments, and adopted the other two land cover products to discover the impact of the discrepancy of land cover on the estimation of BVOCs emission. And in addition, we also used the CCI-LC to do the sensitivity tests of cross-walking table, and the results have been added to the Discussion section in the revised manuscript and the supplement.

The discussion about the impact on BVOC emissions from PFTs were in page 10, line 26 of the revised manuscript as following:

"The BVOC emission results from the MEGAN model are more sensitive to the PFTs than to the LAI. Which LC dataset is used in the model significantly affects the BVOC emission estimates (Zhao et al., 2016; Wang et al., 2011). There are two major sources of uncertainty in the PFTs: the accuracy of the LC map and the cross-walking table used to convert the LC classes to PFTs (Hartley et al., 2017). 61 sample points in Beijing and the surrounding area from the Land Cover Validation Dataset by Zhao et al. (2014) were used as primary validation for the LC data sets. The validation samples were collected from TM/ETM images for 2009–2011. The accuracies of the FROM, MODIS, and CCI-LC datasets are 59.67%, 54.1%, and 50.81%, respectively. Since FROM LC has the same benchmark period as the validation dataset and similar spatial resolution, the FROM LC displayed better accuracy than the other two products. The validation results only can only coarsely assess the accuracy of the LC data sets. The advantage of the high-resolution data is that it diminishes the uncertainties associated with mixed pixels in the medium-resolution LC map, which relies on the cross-walking table to convert the LC classes to PFTs.

The uncertainties associated with the cross-walking table are more evident in CCI-LC. The cross-walking table used in E5 is the default table designed for the global scale. Therefore, two more sensitivity experiments were designed using the "minimum biomass" (minCW) and "maximum biomass" (maxCW) cross-walking tables provided by Hartley et al. (2017) for CCI-LC to examine the uncertainties associated with the cross-walking table. In the maxCW simulation, the area fractions of broadleaf and needleleaf trees were 29.9% and 8.8% (Table S3), respectively, which are similar to those of FROM and MODIS (Figure S1), while the minCW simulation led to relatively low area fractions of 10.9% and 3.7% for broadleaf trees and needleleaf trees, respectively. The BVOC emission estimates are shown in Table S4. Compared with the results of E5, the maxCW and minCW simulations led to a 48.1% increase and a 44.7% decrease in isoprene and a 20.2% increase and a 33.3% decrease for monoterpene, respectively, indicating the strong effect of the cross-walking table on the BVOC estimates, which is more significant for the medium-resolution map. But for a high-resolution LC map based on TM/ETM images like FROM, high spatial resolution could diminish the uncertainty from cross walking processes. Furthermore, the BVOC emissions in the maxCW experiment are similar to the results of E1 with FROM LC and E4 with MODIS LC: a 7.0% increase for isoprene and 9.0% decrease for monoterpenes compared to E1, and a 1.0% increase for isoprene and 8.6% increase for monoterpenes compared to E4. Examining the features of local forests provided by high resolution LC map shows that the maxCW is likely a more accurate representation of the local PFTs. The BVOC emission estimates in

the Beijing region using the three LC data sets are similar, falling within the ranges of 28.5–30.5 Gg for isoprene, 9.3–10.1 Gg for monoterpenes, 1.2–1.4 for sesquiterpenes, and 28.3–35.6 Gg for other VOCs."

5) Finally, it is evident that the modeling exercise described in this manuscript generally supports the results and conclusions of a similar study by Ren et al. (http://dx.doi.org/10.1016/j.envpol.2017.06.049) for the same region (Beijing) that covers the same topic more thoroughly. The Ren et al. paper is not referenced in this manuscript which is not surprising since it was only recently published. However, it is important that the authors do compare with and discuss the results and conclusions of the Ren et al. paper and consider whether (and how) their manuscript adds any new information to the existing scientific literature.

Response: The authors thank for the reviewer's comments. Ren et al. (2017) presented the similar research about the BVOCs emission in Beijing during 2015. The two studies adopted the similar algorithms but the data sources were different. As mentioned above, Ren et al. (2017) adopted the statistical data of the main tree species and collected thorough parameter of the vegetation, which contains more detail compared with our previous data. Therefore, we adjusted our standard emission factor based on the data from Ren et al. (2017) and recalculated the BVOCs emission. The corresponding results and analysis would be presented in the revised paper. As emphasized by the reviewer, the results of Ren et al. (2017) are more thorough, and the comparison of two studies could be helpful to understand the discrepancy of estimations of MEGAN model and more accurate species-based estimation. The modification about the comparison of two studies were in page 12, line 16 of the revised manuscript as following:

"The estimates made by Ren et al. (2017) used a species-level vegetation inventory based on field surveys, while this study used PFT-scale estimates based on satellite datasets, with statistically-derived PFT emission factors. The former method may be more accurate than the latter since emission factors differ between tree species, but it is limited by the rough process of data collection, and satellite based inventories are more easily gridded, facilitating coupling with chemistry transport models and thus allowing further investigation of the effect of BVOCs on atmospheric chemistry. Moreover, in Ren et al. (2017) the coverage of broadleaf trees and needleleaf trees is 18.7% and 7.4%, respectively, and in this study, it is 20.0%−30.0% and 6.3%−7.3%, respectively; i.e., the coverage of needleleaf trees, the main contributors to the emission of monoterpenes, is similar between the two studies, and thus the monoterpene emissions are relatively consistent. However, the isoprene estimates in this study are generally higher than those of Ren et al. (2017): the ratios of isoprene emissions between the two studies is 1.15−1.7, which are similar to the ratios (1.06−1.6) of broadleaf tree area between the two studies."

And the discussion about the new information conveyed by the study of Ren et al. (2017) was added to page 13, line 8 of revised manuscript as following:

"Ghirardo et al. (2016) and Ren et al. (2017) also investigated BVOC emissions from urban green space in the Beijing region. Considering the strong anthropogenic emissions and anthropogenic forcings such as high temperatures and ozone pollution, BVOC emissions from urban green space may have a more direct and stronger impact on urban air quality than suburban and rural emissions. However, this is difficult to evaluate using a mesoscale model like MEGAN, which relies on satellite-based datasets. Therefore, the field-survey based research discussed above may play an important role in future studies concerning the impact of urban BVOC emissions on air quality."

**To Referee #2:**
**General comment**

1) The authors report the sensitivity of WRF (v3.3.1)-MEGAN (v2.1) calculated BVOCs emissions to land cover and LAI inputs over the Beijing area in 2013. The results were compared with previous studies and related to regional air quality. This material is original and suitable for ACP.

Response: The authors appreciate your precious time and effort to improve the quality of our manuscript. The aim of our studies is to investigate the natural effect on air quality, and the future work will focus on the air quality simulation to quantitatively investigate the natural effect on air pollution. This manuscript is the first step of this topic and we concentrated on the MEGAN model and its sensitivity to different inputs. Considering the comments from two reviewers, we has adjusted the contents of our manuscript, and the data and results from a recent published paper concentrated on same topic by Ren et al. (2017) have been added in the revised manuscript to further discuss this topic.

2) The methodology section needs to be significantly expanded to include more descriptions on experiment setup (see my specific comments below). Any novel settings should be highlighted.

Response: The authors thank for your comments. We have replied the reviewer's specific comments below and will add more details of the configuration of the experiments in the revised manuscript and the supplement.

3) The paper can benefit from careful language editing, preferably with help from a native English speaker. I recognize that the authors made some efforts to address this issue brought up during the ACPD quick report phase. However, the current version still contains many grammar mistakes and awkard sentences. References are often inaccurate/inappropriate. Transitions from one sentence to another, from one paragraph to another are not smooth. Also, when one paragraph ends and a new one begins, the authors should either indent the first line of the new paragraph, or leave a line space between the two paragraphs. Here are some suggested edits to the first sentences of your abstract.

P1, L11: air quality pollution -> air pollution

P1, L12: delete "still", and also requires other emission inventories. A sentence saying BVOC emissions are sensitive to land and met conditions should be placed here.

P1, L15-16: "based on" -> "using"; add "the" before "Model of ....."; delete "model" after v2.1

P1, L19: "are used to design five experiments, as E1-E5, to calculate and test the sensitivity of the model" -> are used in five model sensitivity experiments, as E1-E5

P1, L20: "Based on the meteorological conditions from Weather Forecasting and Research (WRF) model, this inventory is an hourly inventory with 3-km spatial resolution."->These sensitivity calculations were driven by hourly, 3 km meteorological fields from the Weather Forecasting and Research (WRF) model.

Response: The authors thank for your constructive suggestion. The language issue was mentioned by two reviewers and the professional language editing has been called before the revised manuscript submitted.

[Figure]

**ELSEVIER**
**Language Editing Services**
*Registered Office:*
Elsevier Ltd
The Boulevard, Langford Lane,
Kidlington, OX5 1GB, UK.
Registration No. 331566771

**To whom it may concern**

The paper "Sensitivity of Biogenic Volatile Organic Compounds Emissions to Leaf Area Index and Land Cover in Beijing" by Hui Wang, Qizhong Wu was edited by Elsevier Language Editing Services.

Kind regards,

**Elsevier Webshop Support**

The abstract was revised as following:

"**Abstract.** The Beijing area has suffered from severe air pollution in recent years, including ozone pollution in the summer. In addition to the anthropogenic emissions inventory, understanding local ozone pollution requires a reliable biogenic volatile organic compound (BVOC) emission inventory. Forest coverage rose from 20.6% to 35.8% from 1998 to 2013 in Beijing according to the National Forest Resource Survey (NFRS). In this study, we established a high resolution BVOC emission inventory in Beijing using the Model of Emission of Gases and Aerosols from Nature (MEGAN) v2.1 with three independent leaf area index (LAI) products and three independent land cover products. Various combinations of the Global LAnd Surface Satellite (GLASS), Moderate-Resolution Imaging Spectroradiometer (MODIS) MCD15, and GEOland (GEO) v2 LAI datasets and the Finer Resolution Observation and Monitoring of Global Land Cover (FROM-GLC), MODIS MCD12Q1 PFT products, and Climate Change Initiative Land Cover (CCI-LC) products are used in five model sensitivity experiments (E1-E5). These sensitivity calculations were driven by hourly, 3 km meteorological fields from the Weather Research and Forecasting (WRF) model. The following results were obtained: (1) According to the baseline estimate, the total amount of BVOC emissions is 75.9 Gg for the Beijing area, and the isoprene, monoterpenes, sesquiterpenes and other VOCs account for 37.6%, 14.6%, 1.8%, and 46% of the total, respectively. Approximately three-quarters of BVOC emissions occur in the summer. (2) According to the sensitivity experiments, the LAI input does not significantly affect the BVOC emissions. Using MODIS MCD15Q1 and GEO v2 LAI led to slight declines of 2.6% and 1.4%, respectively, of BVOC emissions in the same area. (3) The different PFT inputs strongly influenced the spatial distribution of BVOC emissions, which is determined by the spatial distribution of PFTs. Furthermore, the cross-walking table for converting land cover classes to PFTs also has a strong impact on BVOC emissions; the sensitivity experiments showed that the estimate of BVOC emissions by CCI-LC ranged from 42.1 to 70.2 Gg depending on the cross-walking table used. Adopting the "maximum biomass" table made the CCI-LC relatively consistent with the other two LC products, such that the estimates of BVOC emissions in the Beijing region by the three LC products consistently fell within the range of 28.5¬¬–30.5 Gg for isoprene, 9.3–10.1 Gg for monoterpenes, 1.2–1.4 Gg for sesquiterpenes, and 28.3–35.6 Gg for other BVOCs. The development of forest areas and the active greening policy could be the main driver of this increase by stimulating the growth of BVOC emissions."

**Specific suggestions and comments**

1) Novelty: The authors argue that using spatially and temporally varying meteorological fields output from WRF is advantageous, compared with the approaches in some previous studies. Using WRF fields to drive MEGAN calculations is not at all a novel approach and has been widely used in a large number of studies, including some cited by the authors. As the authors are already aware, the uncertainty in their WRF simulation contributed to the estimated BVOC emission biases. There is no need to emphasize the benefit of driving MEGAN using WRF. Rather, if any novel configurations were applied to your WRF simulation, which helped reduce errors in the modeled T2, radiation, moisture, etc, they should be highlighted. See also my next comment.

Response: The authors appreciate your constructive comments. The authors agree to the reviewer's point of "driving the MEGAN model by mesoscale meteorological model is not a novel approach", indeed, multiple studies have adopted same methodology(Carlton and Baker, 2011;Shuping et al., 2010;Wang et al., 2011;Li and Xie, 2014). In this study, we emphasize this part to explain the different consideration of meteorological conditions compared with some previous studies. Compared with some previous studies (Klinger et al., 2002;Zhihui et al., 2003), this method could be the more reasonable way to explain the meteorological effect on the BVOCs emission. And the accuracy of the meteorological conditions is helpful to diminish the uncertainties of BVOCs emission from meteorological conditions. And we adopted default MODIS land cover provided by WRF-official group in our previous simulation. Therefore, considering reviewer's suggestion, we re-simulated the WRF model with updated land cover by using MODIS 12Q1 data for the summer in 2013. As showed in Table 2, the simulation results were validated by hourly

in-situ temperature observation. The average *R* has a slight increase from 0.82 to 0.83, but the Root-Mean-Square-Error (RMSE) and Mean Error is increased from 2.67 ℃ and 3.34 ℃ to 3.07 ℃ and 3.70 ℃, which means updating land cover of MODIS is not beneficial to improve the model performance under this situation. And the specific sites like Tong Zhou, Da Xing and Fang Shan still have the underestimation of temperature simulation, which could not be solved by the updating the land cover. Since the work mainly focus on the sensitivity of LAI and LC, the more effort would be paid on discussing the effect of LAI and LC inputs.

Table 2. The meteorological validation with in-situ observation. MB, ME and RMSE is the abbreviation of Mean Error, Mean Bias and Root-Mean Square Error, and the *R* is the correlation coefficient.

| Default MODIS land cover in WRF | | | |
|---|---|---|---|
| ID | ME(°C) | MB(°C) | *R* | RMSE(°C) |
| Beijing | 2.07 | 0.68 | 0.83 | 2.76 |
| Hai Dian | 2.16 | 0.93 | 0.83 | 2.93 |
| Chao Yang | 2.09 | -0.48 | 0.82 | 2.67 |
| Shun Yi | 2.02 | -0.35 | 0.84 | 2.61 |
| Huai Rou | 2.13 | 1.4 | 0.84 | 2.95 |
| Tong Zhou | 4.26 | -4.03 | 0.78 | 4.9 |
| Chang Ping | 2.3 | 0.75 | 0.79 | 3.19 |
| Yan Qin | 3.7 | 3.36 | 0.79 | 4.64 |
| Feng Tai | 2.31 | -1.19 | 0.82 | 2.88 |
| Shijing Shan | 2.09 | -0.25 | 0.81 | 2.68 |
| Da Xing | 4.53 | -4.42 | 0.81 | 5.12 |
| Fang Shan | 3.9 | -3.63 | 0.79 | 4.48 |
| Mi Yun | 1.94 | 0.46 | 0.83 | 2.52 |
| Mentou Gou | 1.92 | 0.44 | 0.83 | 2.6 |
| Ping Gu | 2.57 | -1.77 | 0.85 | 3.1 |
| Avg | ***2.67*** | -0.54 | 0.82 | ***3.34*** |
| MODIS 12Q1 land cover in 2013 | | | |
| ID | MB(°C) | ME(°C) | *R* | RMSE(°C) |
| Beijing | 2.93 | 2.64 | 0.85 | 3.59 |
| Hai Dian | 3.23 | 2.97 | 0.83 | 3.99 |
| Chao Yang | 1.96 | 0.02 | 0.83 | 2.53 |
| Shun Yi | 1.87 | -0.22 | 0.85 | 2.41 |
| Huai Rou | 3.67 | 3.56 | 0.83 | 4.39 |
| Tong Zhou | 4.14 | -3.92 | 0.8 | 4.73 |
| Chang Ping | 2.89 | 2.54 | 0.81 | 3.8 |
| Yan Qin | 5.34 | 5.33 | 0.8 | 6.04 |
| Feng Tai | 2.15 | -0.94 | 0.83 | 2.68 |
| Shijing Shan | 2.35 | 1.39 | 0.82 | 2.97 |
| Da Xing | 4.47 | -4.38 | 0.82 | 5.01 |
| Fang Shan | 3.89 | -3.67 | 0.8 | 4.46 |
| Mi Yun | 1.93 | 0.78 | 0.85 | 2.48 |
| Mentou Gou | 2.67 | 2.14 | 0.82 | 3.36 |
| Ping Gu | 2.6 | -1.92 | 0.85 | 3.13 |

| | | | | |
|-----|------|------|------|------|
| Avg | *3.07* | 0.42 | 0.83 | *3.70* |

2) More information regarding your WRF simulation and evaluation approach should be provided in Section 2.2.1. These should include:

Response: The authors are grateful for your valuable suggestions. As mentioned above, we have supplemented the description about configuration and evaluation of the model in the revised manuscript. And corresponding questions or comments would be replied point to point as following:

*-introduce the initialization time for each domain.*

Response: The model was initialized at 12:00 UTC every day, and initial and boundary conditions were provided by the FNL(Final) Operational Global Analysis data(National Centers for Environmental Prediction, 2000); the boundary conditions were updated every 6 hours. The first 12 hours were treated as the spin-up time. And we cut and merged the medium 24 hours, from 00:00 A.M. UTC to 23:00 P.M. UTC, to drive the MEGAN model to estimate the BVOCs emission.

And the corresponding content were added to page 5, line 8 of revised manuscript as following:

"The WRF model was initialized at 12:00 UTC, and the first 12 hours were spin-up time. The data of the period from 00:00 A.M. UTC to 23:00 P.M. UTC in the second day was cut and merged to estimate the BVOC emissions."

*-introduce the vertical spacing for each domain.*

Response: The three domains in the model all contain same 27 vertical layers above the ground and 4 vertical layers under the ground, and the sigma values of model vertical layers are:

1, 0.993, 0.983, 0.97, 0.954, 0.934, 0.909, 0.88, 0.8295757, 0.7791514,0.7287272, 0.6783029, 0.5917439, 0.5136936, 0.4434539, 0.3803751, 0.3238531, 0.2733261, 0.228273, 0.18821, 0.1526888, 0.1212943,0.09364247, 0.0693781, 0.04817315, 0.02972473, 0.01375316, 0.

And the corresponding content were added to page 5, line 8 of revised manuscript as following:

"The model domain contained three horizontally nested grids with the spatial resolution of 27-9-3 km and 31 layers vertically."

*-introduce your physics options for each domain and their suitability for the Beijing area (based on literature and/or any sensitivity simulations the authors may have conducted)*

Response: The authors appreciate your previous comments. The physical options have been presented in the supplement.

And the corresponding modification were in page 5, line 13 of the revised manuscript as following:

"The physical options used for the WRF model are presented in Table S2 in the supplement."

*-introduce the land cover and vegetation dynamics (e.g., green vegetation fraction) input data, including the year of these input data represented, and discuss how they may have contributed to biases from your WRF simulation.*

Response: The land cover of WRF simulation was using default MODIS land cover datasets. The discussion of impact on meteorological simulation by updating the land cover or vegetation fraction is focused on the WRF simulation, which is out of the scope of this study mainly concentrated on. In addition, we also did the sensitivity simulation to test the impact of updating the land cover by the MODIS 12Q1 land cover data in 2013, and hourly temperature validation indicated that such measure didn't significantly help to improve the meteorological simulation. The further discussion about the optimize physical schemes or parameters may be out of the object of this study, and the meteorological validation has demonstrated the reasonability of our meteorological simulation on the key condition like temperature and radiation.

*- P4, L23: Skamarock et al., 2005 is for WRF version 2. Please cite WRF version 3 documentation.*

Response: We have modified this reference from Skamarock et al. (2005) to Skamarock et al. (2008).

- P4, L27: justify "we considered the second day as the reasonable results", for example, compare the 1-day and 2-day model performance.

Response: As mentioned above, the 2-day simulations of WRF were done day by day, and the model was initialized at 12:00 UTC. The simulation lasted 48 hours and first 12 hours were as the spin-up time. The data of the period from 00:00 A.M. UTC to 23:00 P.M. UTC of the second day was cut and collapsed to estimate the BVOCs emission. The "reasonable" part means the simulation without the spin-up time, but such configuration was not expressed clearly by this sentence. Therefore, we modified this paragraph as following to verify the processing of WRF data:

"The WRF model was initialized at 12:00 UTC, and the first 12 hours were spin-up time. The data of the period from 00:00 A.M. UTC to 23:00 P.M. UTC in the second day was cut and merged to estimate the BVOC emissions. The merged file was processed by the Meteorology-Chemistry Interface Processor (MCIP) (Otte and Pleim, 2010) tool to provide meteorological conditions for MEGAN model."

- P4, L29: explain why daily T2 was evaluated, instead of hourly T2? Change "among" to "within";

Response: Considering the observation of radiation is daily, in order to evaluate the two variables at the same level, daily T2 was used to evaluate the simulation ability of model to daily variance of whole year meteorological conditions. We also did the validation with the hourly T2 as presented in Table 3. The validation results are similar with the previous validation with daily temperature. Therefore, there is no obvious difference to use daily or hourly temperature data to validate the simulation and they all illustrate the reasonability of our meteorological simulation.

Table 3. The meteorological validation with hourly T2 in-situ observation.

| ID | ME(°C) | MB(°C) | R | RMSE(°C) |
|---|---|---|---|---|
| Beijing | 1.9 | -0.13 | 0.97 | 2.48 |
| Hai Dian | 2 | 0.1 | 0.97 | 2.67 |
| Chao Yang | 2.35 | -1.07 | 0.96 | 2.91 |
| Shun Yi | 2.32 | -1.3 | 0.97 | 2.93 |
| Huai Rou | 2.12 | 0.28 | 0.96 | 3.13 |
| Tong Zhou | 4.91 | -4.77 | 0.96 | 5.5 |
| Chang Ping | 2.03 | -0.4 | 0.97 | 2.73 |
| Yan Qin | 3.11 | 1.88 | 0.94 | 4.3 |
| Feng Tai | 2.9 | -2 | 0.96 | 3.55 |
| Shijing Shan | 2.34 | -0.98 | 0.96 | 2.86 |
| Da Xing | 5.22 | -5.04 | 0.95 | 5.92 |
| Fang Shan | 4.73 | -4.44 | 0.94 | 5.6 |
| Mi Yun | 2.61 | -0.36 | 0.94 | 3.66 |
| Mentou Gou | 2.01 | -0.5 | 0.97 | 2.57 |
| Ping Gu | 3.3 | -2.53 | 0.95 | 4.25 |
| Avg | 2.92 | -1.42 | 0.96 | 3.67 |

- P5, L1/L4: unit is missing for these biases. Why was the MB of -1.5 degree mentioned twice?

Response: Thanks for your precious comments. We have followed the reviewer's suggestion and added the unit of biases in the article. The MB mentioned twice is for emphasizing the general cooling bias of the temperature simulation.

Response: It's Beijing Station, No. 54511. The information has been added to the revised manuscript as following:

"The daily downward shortwave radiation (DSW) was also validated using the in-situ data from Beijing Station."

3) Issues regarding satellite products:

- P6, L22: "Because of the highest spatial resolution of the FROM LC product, the experiment using FROM PFT and GLASS LAI as inputs is the baseline experiment (E1)". I don't understand the logical connections between resolution and choice of the baseline experiment.

Response: The authors appreciate your precious comments. According to the description of MEGAN(Guenther et al., 2012), the emission factor is decided by the distribution of Plant Function Types (PFTs). The sub-grid categories in the specific gird are presented by calculating the area fraction of different PFTs, which means the high-resolution can provide more details of PFTs distribution and calculate more accurate fractions of PFTs. Therefore, we treated the Fine Resolution Observation and Monitoring of Global Land Cover (FROM-GLC) land cover datasets with 30m spatial resolution as the baseline experiment.

- Although the land cover datasets used in this study differed by at least a factor of ten in resolution (30m vs 500/300m), they are all at much finer resolution than the 3 km WRF-MEGAN grid. It would be helpful to explain how these data were regridded to your WRF-MEGAN model grid. This would help us understand how the original data resolution may have affected your results. Approach used to reproject the original 1 km LAI data should also be provided. Missing a "respectively" in P5, L25.

Response: The authors agree with the reviewer. The WRF-MEGAN grid is coarser than the land cover grids, and we used the Preprocessing Tools of MEGAN to regrid the grids by calculating the area fractions of different landscapes (or PFTs) in WRF grid. Furthermore, the original LAI data was also regridded to WRF-MEGAN grid through the calculating the area mean LAI.

The corresponding modification were in page 6, line 5 of the revised manuscript as following:

"The LC datasets were regraded to the WRF grid by calculating the area fraction of each PFT, and the LAI datasets were converted from original grids to WRF grids by calculating the area mean LAI in the WRF grids."

- P5, Section 2.2.2: The land cover and LAI data citations are not helpful. For each dataset, please cite the corresponding algorithm/validation paper, and provide the dataset doi or/and accurate links to retrieve the data. MOD15 is not an accurate description for the used MODIS LAI data. I assume the correct format should be MCD15A2H, Collection/Version XX. Same issue exsits in Table 4.

Response: The authors thank your constructive comments. We have cited the relevant papers and added the source of the datasets to help readers to retrieve the data, and the name of MODIS LAI data has been modified. The availability of datasets and code was added as an independent section in revised manuscripts as following:

"The source code of WRF model V3.3.1 and MEGAN v2.1 is available at http://www2.mmm.ucar.edu/wrf/users/ and https://bai.ess.uci.edu, respectively. The FROM-GLC can be downloaded from the website of Department of Earth System Science, Tsinghua University, at http://data.ess.tsinghua.edu.cn/index.html. The CCI-LC can be downloaded from the website of Climate Change Initiative Program at https://www.esa-landcover-cci.org. The GLASS LAI can be obtained through the website of National Earth System Science Data Sharing Infrastructure at http://www.geodata.cn/thematicView/GLASS.html or the website of Global Land Cover Facility, University of Maryland, at http://glcf.umd.edu/data/lai/. The GEO v2 LAI is available on the website of the Copernicus Global Land Service at https://land.copernicus.eu/global/products/. The MODIS MCDQ12 LC and MODIS MCD15A2 LAI, Version 5, are available on the website of Land Process Distributed Active Center at https://lpdaac.usgs.gov/dataset_discovery/modis/modis_products_table."

- Did the author screen the LAI data and if so, based on what criteria? What values were used for grids with missing data/unrealistic (e.g., extremely high) LAI? Previously studies have reported MEGAN sensitivity to PFT and LAI, so it'd be helpful to compare your findings with theirs.

Response: The authors appreciate your comments. In this study, the LAI datasets are the level 4 satellite products, and the GEOv2 as well as GLASS LAI products have adopted some measures to remove and fill the unrealistic or

unreasonable value (Verger et al., 2014;Xiao et al., 2014). The MODIS LAI products adopt vegetation canopy radiation models of diverse plants type to produce LAI products, and if the canopy model is not available for the pixel, the backup algorithm of estimating LAI by using Normalized Difference Vegetation Index (NDVI) would be used (Knyazikhin et al., 1999). And the MODIS products also provided the quality flags to distinguish the quality of products, and the remain missing values are the pixel mixed with no-vegetation types like water. Considering the characters of the MODIS LAI, we checked all the available values to make sure that they are in reasonable range of LAI (0-7), and used all available values. We didn't use interpolating method to fill the missing value to avoid extra uncertainty and further compared the effect of LAI products by considering two aspects, masking area and LAI value discrepancy.

The corresponding discussion about LAI and PFTs were added to page 10, line 17 in the revised manuscript as following:

[revised manuscript text omitted]

5) The uncertainty section (3.5) is not well written, and the current discussions are very qualitative and not informative.

Response: The authors thank for your comments. We have removed this section and added more informative as well as quantitative results in the Discussion section.

6) P2, L25-27: Please provide the source for "the statistical data from the Nation Forest Resources Survey (NFRS) reported that the forest coverage rate in Beijing rose from 20.6% to 35.8% during 1998-2013". This is also the right place to mention the impact of different met conditions during the earlier periods and 2013.

Response: The authors thank for the reviewer's comments. The data of forest coverage rate of China and specific provinces came from the website of the China Forestry Database (http://data.forestry.gov.cn/lysjk). We also followed the met conditions effect on this period at the same position. The corresponding modification is showed as following:

"Previous studies have carried out some calculations of local BVOC emissions (Wang et al., 2003;Klinger et al., 2002), but these estimates were made for an earlier period (1998–2000), and the China Forestry Database (http://data.forestry.gov.cn/lysjk) provided by the National Forest Resources Surveys shows that the forest coverage rate in Beijing rose from 20.6% to 35.8% from 1998 to 2013."

7) P1, L14; P2, L28; P3, L1: "new" is not accurate. Previous estimates of BVOCs emissions introduced by the authors are not for 2013.

Response: Thanks for your comments. We have modified it in the revised paper and removed the word "new".

8) P5, L25: Shouldn't the last sentence belong to Section 2.2.1? Define MCIP, and use the correct link for MCIP.

Response: The authors appreciate your comments. We have move the last sentence of this paragraph to Section 2.2.1, and the abbreviation of MICP was extended to the full name, Meteorology-Chemistry Interface Processer, with citing the corresponding paper from Otte and Pleim (2010).

8) P5, L28: four species -> four groups

Response: Thanks for your precious comments. We have followed the review's comments and modified this error.

9) P9, Section 3.3: Method of this sensitivity test should be first introduced in Section 2.

Response: Thank you for the comments. The introduction of the contribution calculation has been removed considering the comments of the first referee.

10) To comply with the ACP policy, data availability should be included in the "Acknowledgements" section.

Response: Thank you for the comments. We have followed the reviewer's comments and added data availability in the revised manuscript as following:

"The source code of WRF model V3.3.1 and MEGAN v2.1 is available at http://www2.mmm.ucar.edu/wrf/users/ and https://bai.ess.uci.edu, respectively. The FROM-GLC can be downloaded from the website of Department of Earth System Science, Tsinghua University, at http://data.ess.tsinghua.edu.cn/index.html. The CCI-LC can be downloaded from the website of Climate Change Initiative Program at https://www.esa-landcover-cci.org. The GLASS LAI can be obtained through the website of National Earth System Science Data Sharing Infrastructure at http://www.geodata.cn/thematicView/GLASS.html or the website of Global Land Cover Facility, University of Maryland, at http://glcf.umd.edu/data/lai/. The GEO v2 LAI is available on the website of the Copernicus Global Land Service at https://land.copernicus.eu/global/products/. The MODIS MCDQ12 LC and MODIS MCD15A2 LAI, Version 5, are available on the website of Land Process Distributed Active Center at https://lpdaac.usgs.gov/dataset_discovery/modis/modis_products_table."

11) Captions of Figures 4, 6, 7: specify which experiment these were based on.

Response: Thank you for the comments. We would modify these figures and make them more clear and informative.

---

## Author Response (AR3)

**General comment**

The authors quickly made some changes according to my comments. I find the responses to my line-by-line comments are generally satisfactory. To address my major concerns, the authors added additional figures showing time series of temperature and radiation at the Beijing station. They also included detailed discussions or caveats. I find some of these changes can be further improved.

Response: The authors thank for your encourage and sharing your time in this manuscript. The detailed responses to the comments are given point to point.

One of the major limitations of this paper is that the calculated BVOC emissions are not evaluated by ANY sort of observations or observation-constrained emission estimates. I understand that flux measurements may not be available and making top-down estimates from satellites would be very challenging for this region.

Response: The authors appreciate your precious comments. It is true that the lack of canopy-scale observation makes it difficult to qualitatively evaluate the inventory, so what we can do is focusing on the inputs of the model and giving a primary estimation of local BVOCs emission. If it is possible, we would involve some observation research on this topic by cooperating with other individuals in the future.

I also understand that this paper concentrates on LAI and PFT, and perhaps that is why the analysis on WRF performance is not quite emphasized. However, as the authors recognized, the WRF performance is important to BVOC calculations. One way to improve your WRF story is to consider reporting temperature and radiation (absolute values from the model) at more sites, or showing the data in maps by season, even though radiation can only be evaluated carefully at the Beijing station. This would allow the readers to discern if they are reasonable, or determine ranges of BVOC emissions based on their own met data and the information given from this paper.

Response: The authors appreciate your precious comments. We agreed to the reviewer's point that the meteorological conditions are crucial to estimating BVOCs emission. Therefore, we collected the data of four more meteorological sites in Beijing, meanwhile, some missing values for the former sites are also filled this time. In addition, we also found that there was a mistake in the NCL (NCAR Command Language) script used to extract the downward shortwave radiation (DSW) data from the WRF output file, and this mistake would lead to the time dimension malposition of observation and simulation data, which directly affects the verification of DSW simulation. According to the reviewer's suggestion, we re-evaluated the meteorology simulation by using new observation data and we reorganized this part in the manuscript.

Firstly, the verification statistics by season of average hourly temperature at 2 m height (T2) among 19 sites and DSW in Beijing station are showed in Table R1 to provide the results of general evaluation for meteorological condition, and it has been added to manuscript to replace the previous *Table 2*. As showing in Table R1, the simulation of DSW is relative acceptable refer to other studies like Li et al. (2013) and Wang et al. (2011) currently. Additionally, the Figure R1 was updated and kept as *Figure 1* in the manuscript to directly present the time-series of daily T2 and DSW, but the tab of verification statistics above the figure was removed since the Table R1 has presented the detailed verification statistics. We didn't use the hourly T2 to draw Figure 1 because the hourly data are too dense to present the information. Secondly, the verification statistics of specific sites are presented in Table R2, and it would be added to the supplement to provide the evaluation of each site. The results of four new sites, including Shangdian Zi, Xiayun Lin, Zhai Tai and Tanghe Kou, are emphasized by blue color in Table R2, and verifications of these four sites do not show obvious bias. Meanwhile, Figure R2, which is *Figure 2* in the manuscript, is updated to show the locations of all sites and explain that the simulation bias of the suburban sites, including Da Xing, Tong Zhan and Fang Shan, can be expected to have little impact on the estimate of whole BVOC emissions

because of relative low emission potential.

[revised manuscript text omitted]

The authors moved cross-walking table related material to the main text, which is helpful. They state in the Abstract that "Adopting the "maximum biomass" table made the CCI-LC relatively consistent with the other two LC products, such that the estimates of BVOC emissions in the Beijing region by the three LC products consistently fell within the range of 28.5–30.5 Gg for isoprene…..". Similar statements appear in the main text too. Again, I think the authors should be careful asserting that calculations within the range of 28.5–30.5 Gg for isoprene etc are correct/desirable, due to the lack of evaluation on your BVOC emissions.

Response: The authors appreciate your constructive comments. We agree with the reviewer that the lack of direct verification of BVOC emissions directly limited validity of this kind of conclusion. We think that removing such conclusion and just leave the discussion of the influence of Cross-Walking Table could be a more safe or conservative way.